

# Reviews and syntheses: Current perspectives on biosphere research - 2024

Bohn Friedrich J.[1,2], Ana Bastos[3], Romina Martin[4], Anja Rammig[5], Niak Sian Koh[6], Giles B. Sioen[7,8], Bram Buscher[9], Louise Carver[10], Fabrice DeClerck[11], Moritz Drupp[12,13,14,15], Robert Fletcher[9], Matthew Forrest[16], Alexandros Gasparatos[17], Alex Godoy-Faúndez[18], Gregor Hagedorn[19], Martin Hänsel[20], Jessica Hetzer[16], Thomas Hickler[16,21], Cornelia B. Krug[22], Stasja Koot[9,23], Xiuzhen Li[24], Amy Luers[25], Shelby Matevich[9], H. Damon Matthews[26], Ina C. Meier[12], Awaz Mohamed[12], Sungmin O[27], David Obura[28], Ben Orlove[29], Rene Orth[30], Laura Pereira[31], Markus Reichstein[32], Lerato Thakholi[9], Peter Verburg[33], and Yuki Yoshida[34]

[1]Helmholtz Centre for Environmental Research GmbH - UFZ, Leipzig, Germany
[2]BAM! Bock auf Morgen - Nachhaltigkeit Beratung Medien GmbH VE, Berlin, Germany
[3]Institute for Earth System Science and Remote Sensing, Leipzig University, Leipzig, Germany
[4]Stockholm Resilience Centre, Stockholm University, Stockholm, Sweden
[5]TUM School of Life Sciences Weihenstephan, Technische Universität München, Freising, Germany
[6]University of Oxford, Oxford, UK
[7]Future Earth Global Secretariat, Tsukuba, Japan
[8]National Institute for Environmental Studies, Tsukuba, Japan
[9]Wageningen University, Wageningen, The Netherlands
[10]Lancaster University, Lancaster, UK
[11]Alliance of Bioversity & CIAT, Montpellier, France
[12]University of Hamburg, Hamburg, Germany
[13]University of Gothenburg, Göteborg, Sweden
[14]CESifo, München, Germany
[15]Center for Earth System Research and Sustainability, Hamburg, Germany
[16]Senckenberg Biodiversity and Climate Research Center (SBiK-F), Frankfurt, Germany
[17]University of Tokyo, Japan
[18]Sustainability Research Center, Facultad de Ingenieria, Universidad del Desarrollo, Santiago, Chile
[19]Museum für Naturkunde - Leibniz-Institut für Evolutions- und Biodiversitätsforschung (MfN), Berlin, Germany
[20]Institute for Infrastructure and Resource Management, Leipzig University, Leipzig, Germany
[21]Goethe University Frankfurt, Frankfurt, Germany
[22]University of Zurich, Zürich, Switzerland
[23]University of Johannesburg, Johannesburg, South Africa
[24]East China Normal University, Shanghai, China
[25]Microsoft, Redmond, Washington, USA
[26]Concordia University, Montreal, Canada
[27]Ewha Womans University Seoul, South Korea
[28]CORDIO East Africa, Mombasa, Kenya
[29]Columbia University, New York City, USA
[30]University of Freiburg, Freiburg, Germany
[31]University of the Witwatersrand, Johannesburg, South Africa
[32]Max-Planck-Institute for Biogeochemistry, Jena, Germany
[33]Vrije Universiteit Amsterdam, Amsterdam, the Netherlands
[34]Center for Climate Change Adaptation, National Institute for Environmental Studies, Ibaraki, Japan



**Correspondence:** friedrich.bohn@ufz.de (friedrich.bohn@ufz.de)

**Abstract.** This review of recent advances in biosphere research aims to provide information on selected issues related to changes in biodiversity, ecosystem functioning, social and economic interactions with ecosystems, and the impacts of climate change on the biosphere. We highlight advances on nine themes that have been recently published in peer-reviewed journals that are gaining importance in the scientific community and have the potential to guide future actions as well as inspire future research questions. Our focus is on the interactions between climate, biosphere and society, and on strategies to sustain, restore or promote ecosystems and their services. While mitigating climate change is expected to reduce many risks and associated costs, rapid emission reductions are also crucial to secure various co-benefits of ecosystems, such as coastal protection or stabilization of regional hydrological cycles. In this context, conservation measures implemented in cooperation with local actors are key to efficient resource allocation. At the same time, holistic action frameworks at the global level are required to guide and support such efforts.

## 1   Introduction

Life on Earth as currently organized has been under threat for decades as human activities have changes the planet drastically and without precedent in human history (Watson et al., 2019; Ripple et al., 2023; Rockström et al., 2023; Crutzen, 2006; Stubbins et al., 2021; Cowie et al., 2022; Friedlingstein et al., 2023). As we enter uncharted territory, it is critical that we use scientific evidence as a foundation for decision-making, taking into account the interrelationships within the complex Earth system. The science is clear on the need to significantly cut greenhouse gas emissions, halt biodiversity loss, reduce chemical pollution, and manage ecosystems sustainably to ensure a livable planet (Hill, 2020; Jaureguiberry et al., 2022; Meinshausen et al., 2022). The intertwined crises of climate change and biodiversity loss threatens human well-being, as both crises impact nature processes that support life quality, livelihoods, and economies (Pörtner et al., 2021b, 2023). Our economies are embedded within nature; there is growing recognition from governments and business actors that our economies need to account fully for impacts on nature and rebalance our demands within Nature's capacity to supply (Dasgupta and Treasury, 2022; TNDF, 2023). A whole-of-society approach is needed, as scholars also highlight how fair and just transformations are crucial to reach global climate and biodiversity goals for sustainability and ensuring well-being through sustainable lifestyle and resource circulation practices across food, energy, and material systems (Griggs et al., 2013; Leach et al., 2018; Martin et al., 2020; Folke et al., 2021; Pickering et al., 2022; Obura et al., 2023; McDermott et al., 2023; Schlesier et al., 2024)

At the heart of international negotiations such as the United Nations Framework Convention on Climate Change (UNFCCC) and the Convention on Biological Diversity (CBD), the Intergovernmental Panel on Climate Change (IPCC) and the Intergovernmental Science-Policy Platform on Biodiversity and Ecosystem Services (IPBES) assess the scientific basis for action.



Through regular, comprehensive assessments of the scientific literture (e.g., IPBES, 2019; IPCC, 2021, 2022a, 2023), these bodies provide grounded insights into the current state of knowledge. Their reports inform stakeholders and decision makers about the scientific understanding of climate change and biodiversity loss, its impacts, risks and solutions, and the progress of climate action under international pledges and agreements.

Given the thematic breadth and procedural requirements, IPCC and IPBES assessments take several years to complete. For
example, more than eight years elapsed between the publication of the IPCC AR5 and AR6 Synthesis Reports (Pachauri et al., 2014; Lee et al., 2023) . The first global IPBES assessment report was published in 2019 (IPBES, 2019), and the second global assessment report is scheduled to be completed in 2028. In addition, major reports provide scientific insights with a considerable time lag. For example, the AR6 Synthesis Report was published in 2023, but the cut-off date for the scientific literature reviewed by the three working groups was more than two years earlier, excluding recent publications even in the
year of the report's publication. A limitation of this arrangement is therefore that during the multi-year intervals between these major global reports, negotiators and decision-makers lack an authoritative source for the most recent scientific advances relevant for decision-making. Science-policy interfaces need therefore to develop and improve workflows and mechanisms that allow for rapid deployment of the latest scientific evidence to support policy and decision-making without compromising scientific quality and rigor.

Reports on different aspects of climate change are regularly published such as the IPCC Special Reports, the State of the Global Climate and the Global Carbon Budget (e.g., Pörtner et al., 2019; Organization , WMO; Le Quéré et al., 2013; Friedlingstein et al., 2023) and more recently the State of Wildfires (Jones et al., 2024). Similarly, IPBES special reports and FAO publications such as the State of the World's Forests and the State of Agricultural Commodity Markets (e.g., IPBES, 2023; FAO, 2022a, b) report on biodiversity loss and ecosystem services. These well-recognised reports update diagnostic
indicators familiar to those involved in or following corresponding negotiations. The "10 New Insights in Climate Science" reports address many of the challenges mentioned above, focusing on new findings from recent climate-related research (Martin et al., 2022; Bustamante et al., 2023).

Given the lack of integrative reports on the biosphere, the present publication summarizes recent advances in biosphere research, taking into account social and economic contexts and perspectives. In doing so, it crosses the boundaries of the es-
tablished sciences to provide an interdisciplinary view of biosphere research and to highlight important linkages. This international collaboration aims to inform stakeholders and decision-makers about the latest policy-relevant, peer-reviewed research. We further hope that it may inspire scientists to develop interdisciplinary questions and holistic solutions to pressing problems.

Such Evidence-based solutions using all sources of knowledge are necessary to enable the transformation of socio-environmental systems to conserving ecosystems and enhancing biodiversity, building resilience in socio-ecological systems, restoring de-
graded ecosystems, and promoting a circular and regenerative economy (Chapin et al., 2010; Mace et al., 2018). In this process, it is key to address the main drivers and pressures of environmental degradation, including the conversion and exploitation of biodiversity and ecosystems, climate change, and pollution, as well as divestment from fossil fuels (IPBES, 2019).

Here, we present nine topics with recent and significant findings. To be considered as "new" findings, these advances must be supported by peer-reviewed literature published after 2021 and up to the date of submission. Our topics present impacts



on the biosphere, strategies for maintaining vivid ecosystems or enhancing degraded ecosystems and their services to human society. In addition, we consider emerging themes and research questions that are gaining traction in the scientific community, as well as important future research questions.

We find that biodiversity loss, land degradation, chemical pollution, alteration of biogeochemical cycles and climate change are intricately interlinked across the biosphere and are simultaneously influenced by social and economic systems. Therefore

each topic not only highlights key findings within itself, but also makes connections with related topics to develop a holistic view of changes in biosphere processes and biosphere-human interactions. With this, we aim to identify synergistic approaches to address the complex challenges we are facing.

We note that threats to coastal habitats (Section 3.1), changes in the hydrological cycle (Section 3.2) due to changes in forest cover and shifts in fire regimes (Section 3.3) pose significant societal challenges that require transboundary cooperation

for efficient and equitable resource allocation and distribution. Although climate change mitigation is expected to reduce many of these risks and associated costs, the focus should be on rapidly reducing emissions and ensuring co-benefits, as the effectiveness of natural carbon sequestration (Section 3.4) is likely to be limited by climate change. In this context, adequate conservation measures in human-altered landscapes are a key to maintain nature's contribution to humanity (Section 3.5). At the international level, interlinked and comprehensive policy packages are needed to address the drivers of environmental

degradation from resource extraction (Section 3.6), while at the local and regional level, convivial conservation is a strategy for coexisting with biodiversity within planetary boundaries (Section 3.7).In the future, the socio-economic value of ecosystems will increase with rising real market incomes and changing ecosystem scarcity (Section 3.8). Ensuring societal support and the economic viability of solutions will therefore require a comprehensive change or development of existing nature valuation systems. Finally, we provide an overview of frameworks to guide future action (Section 3.9) that promote equitable, holistic

human-nature relationships and enable a sustainable, inspiring and fruitful future for both people and nature.

## 2  Method

We followed a similar methodology to that of the "10 new insights in climate change" (Martin et al., 2022). First, we set up an editorial board of experts from different fields of ecology, sociology and economics. Meanwhile, we issued an open call inviting the scientific community to submit thematic proposals for this review based on peer-reviewed publications not older

than January 2022. The call for proposals (see Annex A) was disseminated through social media, mailing lists and individual invitations. Despite our efforts to achieve global outreach, we anticipate that we may not have reached some important groups or that they may have chosen not to respond. Hence this first synthesis has to be seen as preliminary effort with caveats that can be improved in next iterations. We expect that this can be a first step towards future annual synthesis reports that will evolve into more substantial, broader-reaching assessments, with a larger pool of input from a more diverse and globally distributed

group of researchers.





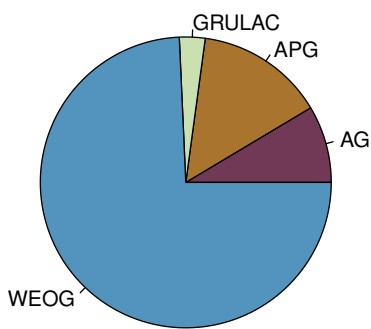

**Figure 1.** Origin of the authors from the geopolitical regional groups of member states of the United Nations: African Group (AG); Asia and the Pacific Group (APG), Latin American and Caribbean Group (GRULAC), Western European and Others Group (WEOG)

We received initially a total of 20 topic proposals. The final selection of topics was made by the editorial board on the basis of the following criteria: (i) sufficient evidence from peer-reviewed publications in the last two years; (ii) emerging general consensus; (iii) relevance to international negotiations and decision-making processes.

The editorial board decision process consisted of two steps. First, each member independently rated the proposals on a scale of 0 to 10, with 0 being 'not recommended' and 10 being 'highly recommended'. The issues were then discussed in a virtual meeting, starting with the highest-rated proposals, with individual ratings adjusted on the basis of the discussion.

Each topic was written by a team of two to five experts selected by the editorial board on the basis of their scientific expertise, as evidenced by their recent scientific publications. Diversity in terms of gender, geography and scientific discipline was also considered (Figure 1, Table 1).

## 3 Insights

### 3.1 Innovative and inclusive solutions offer opportunities to support coastal habitats under threat

#### 3.1.1 Background

Coastal habitats mainly refer to mangroves, saltmarshes, seagrass beds and coral reefs, which are important ecosystems that provide resilience services such as fisheries that contribute to human wellbeing (Costanza et al., 2014; Trégarot et al., 2024).



**Table 1.** Web of Science research areas represented by the authors.

| research area | ∑ | research area | ∑ | research area | ∑ |
|---|---|---|---|---|---|
| environmental sciences | 23 | biodiversity conservation | 13 | ecology | 12 |
| social science, interdisciplinary | 9 | geography | 6 | meteorology, atmospheric sciences | 5 |
| geosciences, multidisciplinary | 4 | remote sensing | 4 | agriculture, multidisciplinary | 3 |
| forestry | 3 | agricultural economics & policy | 2 | anthropology | 2 |
| computer science, interdis. appl. | 2 | economics | 2 | environmental studies | 2 |
| plant sciences | 2 | biology | 1 | cultural studies | 1 |
| engineering, multidisciplinary | 1 | ethics | 1 | marine & freshwater biology | 1 |
| mathematics, interdis. appl. | 1 | physics, applied | 1 | planning & development | 1 |
| political science | 1 | social issues | 1 | urban studies | 1 |

Coastal habitats are important for marine biodiversity (Trégarot et al., 2024) as they function as breeding grounds for fish (Nodo et al., 2023) and shelter for water birds, sequestering carbon at much greater rate than terrestrial ecosystems, and preventing coastal erosion which protects human settlements.

### 3.1.2 Challenges

The importance of a healthy coastal habitat is well established (NOAA, 2024), yet coastal ecosystems are under threat at concerning rates from unsustainable development and climate change (Change , IPCC). For example, 35% of mangroves have been lost because of local drivers but 50% of mangrove ecosystems are at risk of collapse because of climate change and local factors (Hagger et al., 2022). Widespread retreat of coastal habitat is likely at warming levels above 1.5°C (Saintilan et al., 2023). 500 million people are projected to experience challenges within decades due to the likely loss and degradation of coral reefs that they currently rely on (Hoegh-Guldberg et al., 2017). Global warming of 1.5°C to 2.0°C would double the area of tidal marsh exposed to 4 mm/yr of rising sea level by the end of this century. With 3°C of warming, nearly all the world's mangrove forests and coral reef islands and almost 40% of mapped tidal marshes are estimated to be affected (Saintilan et al., 2023). Yet, each coastal habitat responds differently to climate change (Trégarot et al., 2024), making it important to consider local responses. The pressure on coastal habitats from climate change accumulates on top of other anthropogenic stressors such as overtourism, invasive species (Roy et al., 2024), land reclamation (Yamano et al., 2007), pollution (Wakwella et al., 2023), aquaculture, and development of hard infrastructure.

Research on nature-based solutions demonstrate the co-benefits of biodiversity compared to engineered solutions with hard infrastructure (Hahn et al., 2023). This potential means that investing in the space to preserve and recover coastal habitats can help restore biodiversity and mitigate and even help to adapt to climate change but also provide leisurely functions or a source of livelihood. Doing so would improve resilience to a variety of hazards and restore a healthy environment (Hahn et al.,





2023). Moreover, many stakeholders already prefer nature-based over gray infrastructure (Apine and Stojanovic, 2024). While directly beneficial on the local scale, field measurements of over 370 restoration sites in various parts of the world showed that mangrove reforestation provides 60% more blue carbon benefit than afforestation on marginal tidal flats for the same area (Song et al., 2023), suggesting they play an important role for mitigation globally. Utilizing the right mangrove species for the right location may further prevent retreat of the coastal zones, reduce impacts from storms on human settlements, and

positively contribute to fishing grounds among other expected co-benefits (Sunkur et al., 2023). Similarly, recent studies point to the potential of coral reef restoration, combined with coral adaptation and climate change mitigation, to hold off mass coral deterioration and enable reefs to keep up with sea level rise of low to moderate carbon emissions scenarios (Toth et al., 2023; Webb et al., 2023).

Nature-based solutions should be considered with locally relevant species. For example, China introduced an invasive species

called *Spartina alterniflora* (saltmarsh cordgrass) from the USA to reduce soil erosion and provide a number of other ecosystem services in 1979. While successful in fulfilling its purpose, it is occupying the niche of some local plant species and degrading habitat for some water bird species (Nie et al., 2023). Managing invasive species like *Spartina alterniflora* can be costly and complex. Wise use of the biomass may contribute to the local economy, prevent coastal erosion, while still benefit wildlife that depends on them.

**3.1.3   Offering solutions**

Mitigation of coastal habitat loss/degradation can be realized through management and restoration. Trade-offs and synergies between biodiversity conservation/restoration and other services such as carbon sequestration, coastal protection, water purification, aquaculture and eco-tourism should be holistically considered.

Community engagement in restoration of coastal habitats can strengthen willingness to engage in stewardship activities

(Dean et al.) as a result improving biodiversity and climate mitigation outcomes. As demonstrated by the nascent concept of "blue justice" that protests the marginalization of small-scale fishers (Isaacs, 2019), coastal stakeholders (incl. communities, Indigenous peoples, and small-scale fishers) have tended to be excluded from marine decision making (Blythe et al., 2023) yet meaningful community engagement in projects can result in equitable and resilient project outcomes (Fox et al., 2023). Better allowing space for stewardship practices by Indigenous and local communities can provide meaningful lessons for societies

across borders by ensuring livelihoods and biodiversity are restored or conserved (e.g. in California USA, Sanchez et al., 2023). See also section 3.7 & 3.9.

New practices of restoring coastal habitats with co-benefits for people and nature have also been documented (e.g. nature reserve Zwin that consists of dunes, marshes and mudflats along the Belgian and Netherlands border open to tourists and the Mai-po Wetland in Hong kong managed for the benefit of migrating birds, aquaculture and tourism (Cheung, 2011)).

Institutional mechanisms must align to enable innovative or unconventional practices. Institutional barriers to nature-based solutions are currently higher than for gray infrastructure (Jones and Pippin, 2022). Structural recognition of co-benefits of nature-based solutions (Apine and Stojanovic, 2024) could include project funding schemes that recognize the multiple benefits of restoring coastal habitats (e.g. beyond mitigating flood risks), incorporation of feedback from engaged stakeholders into





the project design, and robust monitoring beyond the implementation phase (Palinkas et al., 2022). Researchers have also
begun exploring the role of art in raising awareness around coastal sustainability (Matias et al., 2023). Coastal habitats are
inseparable from upstream land-based activities. Integrated watershed management that transcend jurisdictional boundaries
including through financing for long-term action can foster healthy coastal ecosystems (Wakwella et al., 2023). See also
section 3.6.

### 3.1.4  Recommendations

– Coastal habitat restoration, adaptation, and mitigation efforts should consider the multi-functions of coastal ecosystems.
     Combined gray-green bank protection is recommended for developed coastal zones to strengthen the seawall and provide
     habitat for wildlives.

   – Local species should be prioritized when vegetation re-establishment efforts are being planned to ensure greater co-
     benefits (e.g. when using mangrove or saltmarsh).

– Provide space for ecosystem stewardship. Doing so can benefit coastal habitats and communities.

   – Ensure equitable coastal community decision making through engagement in projects and including co-production prac-
     tices. Wisdom from local and relevant indigenous communities should be adopted when restoring coastal habitats such
     as wetlands (e.g. in the case of aquaculture ponds).

   – Ensure sustainable development upstream using a watershed approach to protect coastal habitats (e.g. preventing nutrient
enrichment, coastal development, hydrologic disturbances, anchoring or sedimentation (Trégarot et al., 2024).

   – Prioritize mangrove reforestation when designing nature-based solutions for mitigating global climate change (Song
     et al., 2023).

### 3.2  Forest protection avoids worsening future droughts and keeps regional, seasonal rain patterns stable

### 3.2.1  Background

Climate change is altering rainfall patterns and intensity in the tropics (IPCC, 2012, 2022b) with implications for ecological and
human water security. Shifts towards more intense rain events, coupled with longer dry spells, potentially leading to increased
incidence of both floods and droughts, have been documented (e.g., Robinson et al., 2021; IPCC, 2023). Also, changes in
the seasonal variability in rainfall patterns across the tropics have been observed (Feng et al., 2013). Fu et al. (2013; 2015)
showed a pronounced shift in dry season length and end due to climate change. In this context, tropical forests can play a
mitigating role because they are strongly coupled to the atmosphere, particularly through the water cycle (Bonan, 2008). The
water cycle in the tropics is driven by interconnected processes, namely evapotranspiration, condensation, rainfall, and runoff.
Each of these components plays a vital role in the health of tropical ecosystems, their ability to support biodiversity and their
capacity to maintain regional rainfall (e.g., Makarieva and Gorshkov, 2007; van der Ent et al., 2010; Spracklen et al., 2012).



High rates of evapotranspiration occur across the tropics due to a combination of intense radiation, large evaporation surface
(the leaves, up to 10 m$^2$ leaves /m$^2$ ground) and high temperatures and significantly contribute to atmospheric moisture. For
example, several studies show that about one-third of the moisture in the Amazon Basin is recycled regionally, while about half
of the moisture in the Congo Basin is recycled regionally (Sorí et al., 2017; Staal et al., 2018; Tuinenburg et al., 2020). This
contributes to cloud formation and generation of rainfall patterns and other regional climatic conditions intricately linked to
forest cover (e.g., Poveda and Mesa, 1997; Ellison et al., 2017). The role can be illustrated by the amount of energy associated
with the evaporation from the Amazon basin, which is greater than 10000 EJ per year and thus more than 10 times higher than
all human energy consumption (fossil, nuclear, and other sources).

Vegetation greening has primarily and increasingly promoted a multi-decadal increase in global ET since the 1980s (Yang et
al 2023). In South America, evaporated water is transported further across the continent contributing to regional rainfall (e.g.,
Zemp et al., 2014, 2017). In some regions, this rainfall provides a large fraction of the water needed for rainfed agriculture
(e.g., Zemp et al., 2014, 2017).

### 3.2.2 Challenges

Despite efforts to curb deforestation, tropical forest loss has accelerated over the last two decades (Feng et al., 2022). Several
lines of research suggest that deforestation reduces regional and downwind rainfall, highlighting again the role of forests in
sustaining regional hydrological cycles (Spracklen and Garcia-Carreras, 2015; Leite-Filho et al., 2021; Staal et al., 2023). Loss
of forest cover disrupts transpiration and reduces precipitation, leading to a drier climate, reduced agricultural productivity and
increased stream flow in large watersheds (Zhang et al., 2017; Zhang and Wei, 2021). In the Amazon basin, this has led to a
measurable decrease in precipitation across South America (Lawrence and Vandecar, 2015). Across the whole tropics, a 1%
reduction in forest cover is thought to have reduced precipitation by an average of $0.25 \pm 0.1$ mm per month over the past
two decades (Smith et al., 2023). Deforestation in South America might delay the onset of the rainy season by 30-40 days
compared to historical periods up to mid-century (Commar et al., 2023; Bochow and Boers, 2023). Modelling studies indicate
that future deforestation in the Congo can reduce local precipitation by 8–10% in 2100 (Smith et al., 2023), and current Earth
system models are known to underestimate recycling in the tropical forests, especially in the Amazon (Baker and Spracklen,
2022). In this context, evidence is mounting that the coupling between the water cycle and vegetation is tightening in many
regions across the globe such that LAI affects ET more strongly over time (Forzieri et al., 2020), and LAI gets more sensitive
to soil moisture availability (Li et al., 2022). However, such stronger water-vegetation coupling is not observed in the tropics
so far. This suggests that tropical forests may be key elements to buffer changes which are already observed elsewhere.

Also links between drought and deforestation have been established (Staal et al., 2020). Here it is assumed that droughts
may be intensified during heatwaves and propagate via teleconnections (Miralles et al., 2019). Droughts have recently more
frequently been observed in many tropical regions. In the Amazon, severe and exceptional droughts occurred in 2005, 2010,
2015 and 2023 (e.g., Jiménez-Muñoz et al., 2016; Papastefanou et al., 2022) with impacts on human well-being in the affected
regions. Also, tropical rainforests were affected (Phillips et al., 2009; Lewis et al., 2011; Tao et al., 2022), which could in turn
lead to forest loss and with this to a reduction in precipitation (Zemp et al., 2017; Bochow and Boers, 2023).



A related challenge is the uncertainty of both observation- and model-based analyses of water and vegetation interactions in the tropics. This arises from the scarcity of soil observations in these regions and the inefficacy of estimating soil moisture

or evapotranspiration using remote sensing techniques, due to dense vegetation. Therefore, also machine learning derived hydrological datasets which extrapolate water variables in space (O. and Orth, 2021; Nelson et al., 2024) are limited in the tropics. In summary, human activities, particularly anthropogenic climate change and deforestation for agriculture and urban development, have profound impacts on the tropical water cycle.It is not yet clear when a tipping point will be reached as a result of deforestation, which would lead to severe dieback due to a drier climate (Lovejoy and Nobre, 2018), but the consequences

for the water and carbon cycle would be severe (Lenton et al., 2019).

### 3.2.3 Offering solutions

Increased efforts are needed to conserve and restore forests and other terrestrial ecosystems by 2030, as pledged in the New York Declaration on Forests and the Glasgow Leaders' Declaration on Forests and Land Use (Gasser et al., 2022). Particularly in areas with high deforestation rates (Feng et al., 2022; Partners, 2023).

Restoring degraded and deforested areas worldwide can increase precipitation and thus mitigate the reduction caused by forest loss (see also section 3.6). But this depends on the regional climate and vegetation characteristics ()(Hoek van Dijke et al. 2022). An increase of forest cover increases both precipitation and evapotranspiration; moisture tracking models can reveal the balance between the two in order to infer the local net effect. In the tropics and particularly the Amazon area this effect is clearly positive, i.e. increased forest cover would lead to increased soil water storage which can then buffer droughts

and fires. Reforestation has the greatest effect in the following areas: In the southern and eastern Amazon, reforestation could increase precipitation, which is critical given the risk of climate change-induced drying and a possible tipping point (Zhao et al., 2017). Similarly, reforestation in middle america and SEA (including south China) could largely offset projected drying, and Mediterranean Europe would also benefit from regional reforestation efforts. Further, due to moisture-recycling of forests, reforestation in the south eastern Amazon would increase gross primary productivity (Staal et al., 2023).

Yet, afforestation for carbon capture in savannahs and other naturally tree-poor ecosystems can disrupt local water balances and biodiversity (Veldman et al., 2015; Fernandes et al., 2016). Trees often use more water than grasslands, which can lower the water table and reduce the availability of water for other plants and animals native to these areas. This change can lead to the drying up of wetlands and lesser water flows in streams and rivers (Farley et al., 2005; Lalonde et al., 2024), impacting species that are adapted to specific water regimes. Moreover, planting non-native tree species can alter soil properties and inhibit the

growth of native vegetation, which relies on fire and open sunlight conditions to thrive (see section 3.3). These ecological shifts can diminish the natural resilience of these ecosystems, making them less adaptable to climatic changes and more susceptible to invasive species. Therefore, while afforestation in certain contexts can be beneficial for carbon sequestration and local societies, it requires careful planning and management to avoid unintended ecological consequences (Farley et al., 2005).

More and more accurate data on tropical vegetation and water could be collected through (i) more standardized and regionally

distributed ground-based measurements and monitoring, and (ii) future satellite missions using longer wavelengths such as SAR L-band (Lal et al., 2023) or p-band missions (Garrison et al., 2024), although the use of the latter is restricted by the





military in many areas. This can provide a basis for more accurate observation-based analysis, and better constrain state-of-the-art models to better quantify the large-scale pan-tropical effect of afforestation or deforestation on the hydrological cycle. Consequently, this can also contribute to a more accurate understanding and estimation of the Amazon tipping point.

### 3.2.4 Recommendations:

- Protecting forests to avoid worsening future droughts and keep seasonal rain patterns stable, particularly in the Congo basin and SEA where high rates of deforestation are feared in the future.

- Reforestation activities needed to increase average precipitation. In particular, reforestation in the southern and eastern Amazon, middle america, SEA (including south China) and Mediterranean Europe would help to increase local precipitation and reduce the severity of droughts.

- Further research is needed to assess reforestation opportunity, major risks and trade-offs (inspired Doelman et al., 2020; Koch and Kaplan, 2022; Yu et al., 2022). Improve monitoring capabilities through more ground measurements in the tropics as often water-related perspective and country or regional level analysis is missing to understand regional-specific feasibility, e.g. country-specific costs and benefits of implementing different reforestation strategies (Griscom et al., 2020)

- Afforestation is even more critical and requires careful planning and management to avoid unintended ecological consequences

### 3.3 Delayed climate change mitigation likely to increase fire risks in many regions

### 3.3.1 Backround

Fire is a natural phenomenon that has shaped many ecosystem types around the world and contributed to their biodiversity (Bond and Keeley, 2005; Pausas and Keeley, 2009; Bowman et al., 2011; He et al., 2019). Humans have altered fire regimes by utilizing fire and changing the landscape, and also suppressing fires to avoid its destructive consequences (Bowman et al., 2011). However, unprecedented record wildfires have recently affected different parts of the world, such as the 18 million ha burn in Canada in 2023 Copernicus (2023), bringing to the fore concerns over future fire dynamics.

Many factors affect fire regimes but recent research suggests that two major factors - human activities (including land use change) and meteorological fire danger - are pulling in opposite directions. On the one hand, human factors (in particular agricultural expansion and intensification in African savannas and grasslands (Andela et al., 2017) have caused a decrease in burned area over the last two decades (Andela et al., 2017; Jones et al., 2022; Chen et al., 2023). On the other hand, increasing fire weather and decreased snow cover have increased burned area and fire intensity in high-latitude regions, e.g. boreal and forests of eastern Siberia and western Canada albeit with large regional variability (Bedia et al., 2015; Jones et al., 2022; Chen et al., 2023; Cunningham et al., 2024; Hessilt et al., 2024). So against a backdrop of globally decreasing fire occurrence, some areas are experiencing increasing extreme fire seasons (Brown et al., 2023; Cunningham et al., 2024) and so-called 'megafires'



which are large, intense and difficult to control (San-Miguel-Ayanz et al., 2013; Collins et al., 2021). These megafires exceed natural fire regimes and are extremely detrimental to biodiversity (Leeuwen et al., 2023), carbon stocks (Clarke et al., 2022; Copernicus, 2023; Zheng et al., 2023), human infrastructure and air quality (Xu et al., 2023).

### 3.3.2 Challenges

Analyses of fire trends and future projections show strong climate-change-induced increases in fire weather severity across most of the world (Abatzoglou et al., 2019; Jones et al., 2022; Jain et al., 2022). This poses a significant challenge for society, particularly forestry and civil protection. However, the problem is highly heterogeneous, with already fire prone areas experiencing increased risk of extreme weather conditions (Scholten et al., 2021; Brown et al., 2023; Cunningham et al., 2024) but also fire prone conditions emerging in relatively cooler and wetter areas that have been little affected by fire so far, e.g. boreal and temperate zones and mountains (de Groot et al., 2013; Jones et al., 2022; Hetzer et al., 2024).

These challenges are heightened by local factors relating to ignitions, vegetation and land cover that can play a major role in increasing fire danger. In some regions, land cover is characterized by highly flammable species such as pine, spruce and eucalypt, and planted in large and homogeneous stands which can promote fire spread. For one of the largest wildfires in central Europe, where Norway spruce monocultures suffer heavily from bark beetle attacks since the exceptional drought in 2018, it has been shown that burn severity was highest in dead spruce stands (Beetz et al., 2024). For fire risk assessments, both climatic and non-climatic factors need, thus, to be considered (European Environment Agency, 2024).

Changing fire regimes also threaten large stores of carbon but with regionally unique consequences. In the humid tropics, intact forest and peatlands are threatened by deforestation fires (Andela et al., 2022; Chen et al., 2023) and wildfires exacerbated by climate and land use change (Turetsky et al., 2015; Harrison et al., 2020). High-latitude peatlands in remote areas are vulnerable to large, long lasting fires burning through deep peat layers (Scholten et al., 2021; Nelson et al., 2021) which are not actively controlled and lead to large carbon losses (Turetsky et al., 2015). Future stocks from potential 'nature-based solutions' may also be vulnerable to wildfires, undermining climate mitigation efforts. However long-term predictions of fire risk that could be incorporated into planning still include large uncertainties at local scale (Hantson et al., 2020). See also Section 3.2 and 3.4.

### 3.3.3 Offering solutions

Decreasing trends in burned area in regions where the fire weather has become more severe, such as non-Mediterranean Europe (Jones et al., 2022) clearly show that fire risks can be mitigated, although at increasing cost (Bayham et al., 2022). The costs of fire mitigation are, however, surpassed by losses, especially for extreme fire seasons (Bayham et al., 2022) and comparable to other climate change mitigation costs (Phillips et al., 2022). Several studies emphasize that the burned area is negatively related to the Human Development Index (Chuvieco et al., 2021; Teixeira et al., 2023). This demonstrates that more economically developed societies are less likely to be severely affected by fires, either due to effective fire prevention measures or because of rapid and successful firefighting (see also Section 3.6).



Strategies should be developed targeting risks at local, national, and regional levels (Chuvieco et al., 2023). Locally, fire suppression can be aided by introducing fire breaks and access points, particularly roads (Haas et al., 2022), however this solution should be applied with caution as land fragmentation also negatively affects species richness (Willmer et al., 2022). Fuel reduction techniques might also be considered, including mechanical or grazing, but prescribed burning might also provide a more natural solution also useful for maintaining fire-dependent vegetation types and biodiversity (Neidermeier et al., 2023). National strategies should promote biodiversity because this also promotes fire resilience by avoiding monocultures of highly flammable species. Furthermore, studies have shown that cross-border collaborations are necessary and effective for allocating resources efficiently and minimizing risk (Bloem et al., 2022). International cooperations can benefit from comprehensive 'fire-smart' solutions, such as those recently targeted in the EU Green Deal (Ascoli et al., 2023; Regos et al., 2023). A number of cases document the value of incorporating Indigenous knowledge and governance into fire management strategies in Latin America (Oliveira et al., 2022), Africa (Croker et al., 2023), North America (Connor et al., 2022), and Australia (Legge et al., 2023). See also Sections 3.6, 3.7, and 3.9.

### 3.3.4 Recommendations:

1. Prepare for more extreme fire weather: To effectively address the upwards trend in fire weather, it is essential to mitigate and prepare for extreme events even in areas not historically heavily affected by wildfires. A smart and strategic use of prescribed burning (and other management practices close to natural processes) could for instance decrease the risk for extreme fires and promote biodiversity. Strengthening firefighting resources by increasing funding for equipment and personnel provides opportunities for fire prevention, surveillance and fire fighting capabilities.

2. Engagement and Collaboration: Raising public awareness through education campaigns and building collaboration with Indigenous Peoples, local communities, and stakeholders ensures a comprehensive and inclusive approach to addressing challenges.

3. Improve fire assessment and prediction: fire risk assessment and forecast needs to consider regional factors beyond fire weather, namely landscape and vegetation properties, management activities, ignitions and socio-economic factors that influence danger, vulnerability and exposure.

4. Consider fire risks in nature-based climate solutions: Nature-based solutions for climate mitigation must holistically incorporate fire risk into their planning. This involves not only assessing the current fire hazard but also anticipating future changes in fire risk.

5. Mitigate climate change: Finally, addressing the root cause of increasing wildfire risks —climate change— is imperative. Rather than merely attempting to manage the symptoms, it is crucial to mitigate climate change to avoid the most extreme, costly and challenging conditions to adapt to.





## 3.4 Nature-based CDR implementation risks

### 3.4.1 Backround

A key intersection point between ecology and climate change research is the ability of terrestrial ecosystems to remove carbon from the atmosphere and store it in above and below-ground carbon stocks. Terrestrial ecosystems currently absorb about a third of total anthropogenic (fossil fuel + land-use) $CO_2$ emissions, mostly as a result of $CO_2$ fertilization of vegetation growth (Friedlingstein et al., 2023). In addition, regrowth of previously deforested land accounts for an additional removal of CO2 from the atmosphere; this regrowth is equivalent to about a third of gross land-use-related CO2 emissions (Friedlingstein et al., 2023).

Given the key role of the terrestrial biosphere as a net carbon sink, there is considerable interest in pursuing strategies to enhance nature-based carbon dioxide removal (CDR) as a contribution to climate mitigation efforts. Many studies have highlighted the potential of nature-based CDR (Griscom et al., 2017; Fuhrman et al., 2023). Reforestation and afforestation are typically seen as the largest potential contributors, though nature-based solutions also include strategies such as biochar and other agricultural management practices to increase soil carbon sequestration. Many concerns about nature-based carbon removal have also been raised in recent literature however, including whether a focus on CDR in research and policy discussion could lead to delays in fossil fuel emissions reductions (Carton et al., 2023), as well as whether nature-based CDR has a large enough potential to be a meaningful contribution to climate mitigation goals (Roebroek et al., 2023). Parr et al. (2024) also highlight an important concern that reforestation with non-native tree plantation species could lead to loss of native ecosystems that may negate any carbon-related gains, supporting previous findings that more biodiverse forests are better at capturing and storing carbon (Liu et al., 2018b). These and other concerns highlight a growing understanding that nature-based CDR must be undertaken with attention to local ecosystems and community needs (Seddon, 2022), and that nature-based CDR should in all cases be treated as a complement (and not an alternative) to fossil fuel $CO_2$ emissions reductions Matthews et al. (2022).

### 3.4.2 Challenges

A key challenge with nature-based carbon storage is that land carbon pools are vulnerable to disturbances, either from natural processes (such as fire) or human pressures (such as deforestation) (Bustamante et al., 2023, see also Section 3.2 and 3.3), as discussed in Section 3.3. Furthermore, climate-driven changes to wildfire and other natural disturbance regimes have the potential to lead to increased vulnerability of land-based carbon stocks with continuing climate change (Anderegg et al., 2020). The potential for land-based carbon storage to be temporary evokes an accounting challenge when used as an offset for fossil fuel $CO_2$ emissions which represent a permanent transfer of new carbon from a geologic reservoir to the atmosphere-land-ocean carbon system. Concerns of impermanence (also referred to as durability concerns or risks of reversal) are a key concern associated with the application of nature-based carbon storage as a contributor to climate mitigation efforts (Zickfeld et al., 2023). However, even temporary carbon storage does have climate value, and in particular has been shown to decrease peak warming if coupled with ambitious fossil fuel emissions reductions (Matthews et al., 2022).





### 3.4.3 Offering Solutions

One solution to this challenge may be to treat all nature-based carbon removal and storage as a temporary quantity and to explicitly account for the amount of time the carbon remains in storage as part of its climate value. Matthews et al. (2023)showed that

carbon storage measured in tonne-years (which represent the time-integral of carbon storage) is proportional to degree-years of avoided warming (the time-integral of avoided warming). Previous applications of tonne-year accounting have focused on trying to equate temporary and permanent storage; however, this use of tonne-year accounting is not grounded in any physical climate science relationship, and leads to a false equivalency of temporary and permanent storage that could further disconnect carbon offset calculations from the scientific understanding of carbon stocks and flows in natural systems (Brander and

Broekhoff, 2023). However, if reimagined as a metric to simply track nature-based carbon storage over time, tonne-year accounting has the potential to measure the climate effect of temporary carbon storage in a way that is coherent with scientific understanding (Matthews et al., 2023) . Measuring and quantifying the time dimension of nature-based carbon storage would require a substantial rethink of current carbon offset protocols, which currently struggle with how to use carbon buffer amounts to guard against loss from disturbance (Haya et al., 2023). An accounting framework based on tonne-year as a proxy for

degree-years of avoided warming has the potential to robustly measure the climate value of temporary carbon storage.

### 3.4.4 Recommendation:

– treat nature-based carbon removal and storage as a temporary quantity

## 3.5 Sustaining Nature's Contributions to People in human-modified landscapes requires at least 20%–25% (semi-)natural habitat per square kilometer

### 3.5.1 Background

Biodiversity is declining faster than ever despite decades of increased conservation investment to bend the curve of biodiversity decline (Leclère et al., 2020). This decline is mainly driven by land and sea use change, resource overexploitation, pollution, exotic species invasions, and climate change (IPBES, 2019). Such decline is also associated with the expansion of global systems of extractivism in recent centuries, contrasting sharply with earlier patterns of stewardship (Ojeda et al., 2022; Molnár

et al., 2024, see also Section 3.6). Converting natural habitats has provided benefits by creating more space for agriculture, housing and industry, but at a significant cost to biodiversity, jeopardizing valuable ecosystem functions and beneficial contributions, such as healthy and sustainable food production, clean air and water, and recreational spaces amongst others. These contributions, known as ecosystem services or Nature's Contributions to People (NCP), directly or indirectly contribute to human well-being, economic stability, and overall quality of life (Díaz et al., 2018, see also Section 3.1 and 3.2 ).

Biodiversity has multiple dimensions making it challenging to define synthetic policy objectives and metrics or track progress (Díaz et al., 2020). Most conservation efforts focus on halting the conversion of remaining intact natural ecosystems, and safeguarding their unique species as articulated in Goal A of the Kunming-Montreal Global Biodiversity Framework (Watson et al., 2018; Allan et al., 2022). However, human-modified lands and waters, covering about half of the global Earth surface





(IPBES, 2019), including highly managed agricultural fields and urban green spaces in mixed mosaic landscapes where natural
functions are limited to small patches of habitat, are often overlooked in conservation policies and global target setting (Pollock
et al., 2020), despite their critical roles in maintaining and supporting human well being and sustainable food production
(Goodness et al., 2016; Díaz et al., 2018). The close proximity and relationship of people with biodiversity in these areas
makes their contributions to human well-being even more important. Identifying metrics to ensure continuous contributions of
such nature to human well-being is challenging due to the highly context-specific conditions under which biodiversity supports
ecosystem functions (e.g. Section 3.2). Yet, few proposals for the post-2020 Global Biodiversity Framework (GBF), address
human-modified lands explicitly or the role of functional biodiversity in maintaining a good quality of life for all people
(Rounsevell et al., 2020; Maron et al., 2021; Hammoud et al., 2024).

NCP provisioning in human-modified landscapes relies on the amount, quality, and spatial arrangement of habitat fragments
and their accessibility to beneficiaries (Garibaldi et al., 2021; Priyadarshana et al., 2024). These landscape components serve
as proxy measures of ecosystem functional integrity (Rockström et al., 2023; Mohamed et al., 2024). Evidence suggests that
many NCP can be maintained by habitat within highly human-modified landscapes as long as a minimum level, quality, and
distance to biodiversity is present, and/or the functional integrity is retained or rebuilt (Martin et al., 2019; Eeraerts, 2023;
Mohamed et al., 2024). The required habitat levels for NCP provisioning vary depending on the context, the NCP, demand for
it, landscape type and taxa involved making it difficult to assess direct relationships (Garibaldi et al., 2011; Cariveau et al.,
2020). Nonetheless, below a certain threshold nature can no longer provide a majority of benefits (Rockström et al., 2023).
A recent systematic review of 154 studies found that the capacity of human-modified lands to pollinate crops, regulate pests
and diseases, maintain clear water, limit soil erosion, and maintain recreation spaces for people significantly declines and often
disappears when habitat area falls below 20%–25% per km2 and nearly disappeared below 10% habitat per $km^2$ (Mohamed
et al., 2024). Alarmingly, only one-third of global human-modified lands are above the 20%-25% per $km^2$ level to sustain NCP
provisioning, emphasizing the urgent need for policy interventions to restore and regenerate ecosystem functions and their
benefits in the remaining two-thirds of global human-modified lands (Mohamed et al., 2024).

### 3.5.2 Challenges

The proposed minimum habitat levels can serve as a general guide to identify priority locations for conservation and restoration
to support sustainable NCP provisions. However, uncertainties remain on the successful implementation of these minimum
habitat levels in practice due to factors such as climate change, habitat loss, unsustainable agriculture, and human settlements
expansion which complicates the implementation and may create trade-offs. General estimates and targets for land management
are important, but often oversimplify the complexities of local conditions and can misrepresent the needs of local communities
due to the inherent biases in ecological research that may not account for all biomes or ecosystem functions (Martin et al.,
2012; Manning, 2024). Additionally, these metrics often overlook finer-scale NCP, e.g., NCP provided by soil biodiversity,
and ignore the important role of complementary agricultural practices such as no-till farming, cover cropping, and leguminous
rotations which can reduce erosion, nutrient loss and maintain biodiversity (Blanco-Canqui et al., 2015; Skaalsveen et al., 2019;
Guinet et al., 2020; Rakotomalala et al., 2023). Current remote-sensing technologies also struggle to detect small and linear





habitat elements or differentiate complex landscape types, likely leading to underestimations of the current state of (semi-)natural habitats globally (Lechner et al., 2009; Jurkus et al., 2022). Therefore, allocating areas to (semi-)natural habitat within
human-modified lands using general estimates, without proper management and consideration of local conditions can conflict with the provisioning of material NCP and might compete with food production ambitions and local community needs (e.g., housing), which is negatively affecting the well-being of local people relying on those NCP (Mohamed et al., 2024).

### 3.5.3   Offering solutions

The implementation of such strategies effectively necessitates adapting and adopting practices that are suited best to local
context and conditions, rather than prescribing a single practice to be applied globally. Countless context-specific strategies exist to enhance NCP provisioning and can be implemented in ways that create more synergies than trade-offs and support food security, livelihood and overall human well-being (Jones et al., 2023; Rakotomalala et al., 2023). For example, modern agroecological practices and nature-based solutions including diverse crop rotations (Shah et al., 2021; Ewert et al., 2023)and mixed cropping systems (Lichtenberg et al., 2017; Tscharntke et al., 2024) maintain habitat heterogeneity and promote ecosys-
tem resilience. Agroforestry systems enhance soil health, water retention, and global carbon sequestration (Zomer et al., 2022; Fahad et al., 2022). Strategically incorporating habitats such as hedgerows, no-mow zones around field margins or other practices (M'Gonigle et al., 2015; Marja et al., 2022; Maskell et al., 2023) combined with innovations such as precision agriculture practices can maintain species diversity (Arroyo-Rodríguez et al., 2020; Knapp et al., 2023) while optimizing agricultural productivity (Balafoutis et al., 2017). Protecting green spaces and parks in cities can enhance physical and mental well-being
(Konijnendijk, 2023) and placing vegetation buffers along waterways can capture sediment and pollutants, among many other tools (Luke et al., 2019).

The 25% high-functioning nature in every square kilometer offers a key policy tool since it is the first widely applicable measurement of the minimum level of human-modified land that needs to be in a (semi-)natural state across several NCP and a wide range of landscapes. This proposed habitat level is the minimum level, not the optimal level required to meet adequate
NCP demand (Mohamed et al., 2024). It serves as a general guideline synergizing with existing policy targets (e.g., UN Decade on Restoration) for prioritizing conservation initiatives and formulating adaptive, scalable policies beyond natural areas. See also Section 3.6, 3.7 and 3.9.

### 3.5.4   Recommendations:

–   Maintain and/or restore at least 20%–25% (semi-)natural habitat per square kilometer in human-modified landscapes to
sustain multiple NCP provisioning targeting agricultural and urban lands.

–   Develop tools and approaches to identify and measure key NCP for any local landscape to determine locally-specific amount (20%-25% per km2), type (composition) and configuration of habitat elements needed for NCP provisioning.

–   Foster, curate and disseminate conservation and sustainable production practices (e.g. agroecological practices) that align with local conditions and community needs and that support ecological and socio-economic outcomes.





– Establish partnerships with Indigenous Peoples, local communities, scientists, and NGOs in decision-making to halt and reverse NCP losses and ensure sustainable conservation efforts since they are the best source of implementable solutions in increasing ecosystem functional integrity.

        – Allocate resources towards innovations in agricultural and other production practices and urban planning, which are biodiversity-friendly (e.g., precision agriculture techniques or the creation of vegetation buffers that support food pro-
490       duction while minimizing the environmental impact.

## 3.6 Interconnect and deliver comprehensive policy packages to address the root causes of degradation caused by resource extraction and to promote sustainable practices

### 3.6.1 Background

Climate change, biodiversity loss, pollution, and land degradation are planetary-scale crises that threaten the sustainability of
our environmental systems (Dasgupta and Treasury, 2022; IPCC, 2023).

        The challenge posed by the relentless extraction of natural resources and the need for a paradigm shift towards sustainable practices require a comprehensive evaluation of conservation strategies, rehabilitation efforts, and the regeneration of depleted ecosystems (Meli et al., 2017; Chazdon et al., 2020). The growing interest from businesses to become 'nature-positive' has highlighted the importance of circular economy principles, which advocate for a systemic transformation that minimizes waste
and promotes the reuse of resources. This approach plays a crucial role in reducing the ecological footprint of resource extraction, aligning business practices with sustainability goals (Bocken et al., 2019; Korhonen et al., 2018; Lüdeke-Freund et al., 2019). Additionally, it explores the intricate nexus between global trade, environmental degradation, and climate change, emphasizing the need for integrated policy packages and international cooperation to mitigate adverse impacts and enhance ecosystem resilience (Leal Filho et al., 2019; IPCC, 2023). Through these measures, we can foster a sustainable future that
balances economic growth with environmental stewardship (Rockström et al., 2017; Steffen et al., 2018). The depletion of natural resources in the Global South demands a shift to sustainable practices that prioritize ecosystem conservation and regeneration (Meli et al., 2017; Chazdon et al., 2020). Circular economy principles, which emphasize minimizing waste and reusing resources, are vital for reducing ecological footprints (Bocken et al., 2019). Additionally, addressing the links between global trade, environmental degradation, and climate change requires integrated policies and international cooperation (Leal Filho
et al., 2019; IPCC, 2023).

        Global trade as it organized today - drives ecosystem vulnerability by imposing environmental externalities that disproportionately affect developing nations with weaker regulations (Newell and Taylor, 2022). In these countries, resource extraction to satisfy global demand often results in severe ecological degradation and social displacement, while economic benefits are enjoyed elsewhere, exacerbating socio-economic inequalities (Barlow et al., 2018; Köhler et al., 2019; Hickel, 2020).This
phenomenon, known as "telecoupling," describes how distant economic activities are interconnected, often resulting in environmental degradation in resource-exporting countries (Liu et al., 2018a). For example, the demand for palm oil in Europe and North America has caused deforestation in Southeast Asia, impacting biodiversity and increasing greenhouse gas emissions



(Meijaard et al., 2020). Similarly, mining for precious metals in Africa to meet industry demands has led to habitat destruction, water pollution, and human rights violations (Northey et al., 2017). Consequently, global trade externalities significantly
contribute to ecosystem vulnerability and social inequity. Therefore understanding global trade networks and their impact is crucial for developing sustainable policies to mitigate these adverse environmental and social effects (Wiedmann and Lenzen, 2018; Wiedmann et al., 2020).

### 3.6.2 Challenges

A special focus on resource extraction highlights the increasing demand for renewable energy sources and the associated need
for minerals and metals. The surge in electric car production, for instance, has driven a significant increase in lithium extraction, particularly in regions like South America's Lithium Triangle. While lithium extraction raises environmental concerns, such as water depletion and landscape disruption, it is generally less harmful than the large-scale extraction of fossil fuels like coal and oil, which have more severe and widespread ecological impacts and contribute significantly to climate change (Vikström et al., 2013; Krishnan and Gopan, 2024). Additionally, the growing bioenergy sector requires extensive land use for biomass
production, which can affect local ecosystems and increase vulnerability to climate hazards such as droughts and floods, exacerbated by climate change (Searchinger et al., 2018).

The concept of telecoupling is crucial in this context, illustrating how Europe's demand for renewable energy technologies can drive resource extraction and associated socio-environmental impacts in regions like the Congo, where materials such as cobalt, which is primarily needed for Lithium-ion-batteries, are sourced (Mancini et al., 2021). This interconnected demand not
only impacts local environments but also influences global climate patterns by shifting where and how resources are extracted and used. For example, deforestation for biomass production in one region can reduce carbon sinks, increasing atmospheric CO2 levels and climate risks globally (see also Section 3.2 and 3.3).

To address these challenges, the adoption of circular economy principles becomes essential. The circular economy focuses on designing out waste and pollution, keeping products and materials in use, and regenerating natural systems. By applying these
principles, we can significantly reduce the need for new resource extraction. For instance, enhancing the recycling of lithium from used batteries can decrease the demand for new lithium mining, thus mitigating its environmental impact (Geissdoerfer et al., 2017). Similarly, recycling and reusing metals like cobalt can reduce the pressures on countries like the Congo, helping to stabilize local ecosystems and communities.

The nexus between resource extraction, environmental degradation, and global trade is complex. International demand accelerates resource extraction, leading to habitat destruction and biodiversity loss as countries exploit their natural assets (Wiedmann and Lenzen, 2018). This extraction often results in pollution, such as water contamination from mining and air pollution from deforestation and fossil fuel combustion, further degrading ecosystems and reducing their functionality.

### 3.6.3 Offering solutions

To increase the resilience of ecosystems and reduce resource extraction and environmental degradation, it is essential to in-
terconnect and deliver comprehensive policy packages. These packages should integrate environmental, economic, and social



policies to address the root causes of degradation and promote sustainable practices. This includes implementing stricter regulations on resource extraction (e.g., Litvinenko et al., 2022), encouraging the adoption of cleaner technologies (e.g., Ikram et al., 2022), and incentivizing the conservation and restoration of ecosystems (e.g., Tedesco et al., 2022; Ostrom, 2009, see also section 3.5). Policies must focus on reducing pollution through improved waste management and stricter emission controls

while addressing climate change by promoting renewable energy sources and enhancing carbon sinks (UNEP, 2022). Recent efforts, such as the European Green Deal, highlight the need for comprehensive policy frameworks that integrate climate action with economic and social goals (Commission, 2019). Additionally, international cooperation is crucial to ensure policies are harmonized across borders, preventing the displacement of environmental harm from one region to another (Mayer, 2018, see also Section 3.7). For instance, the Paris Agreement exemplifies global efforts to align climate policies and reduce carbon

emissions through shared commitments (UNFCCC, 2018). By delivering interconnected policy packages that address multiple dimensions, we can create synergies that enhance ecosystem resilience, support sustainable development, and improve the overall health of the planet (Steffen et al., 2018).

Examples of policy packages that can be deployed to reduce the environmental harmful impacts of trading and drive cooperation between countries include the establishment of international environmental agreements, the implementation of sustainable

trade policies, and the creation of transnational conservation initiatives (see also Section 3.9).

– **International Environmental Agreements:** Agreements like the Paris Agreement set global standards for reducing greenhouse gas emissions and promoting renewable energy sources, encouraging countries to cooperate on climate action (UNFCCC, 2018). Another example is the Convention on Biological Diversity (CBD), which aims to conserve biodiversity, promote sustainable use of its components, and ensure fair and equitable sharing of benefits arising from

genetic resources (CBD, 1992). While these agreements are fundamental to global environmental governance, their tracked record for implementation and achieving targets has been mixed. The CBD, for example, is effective in reporting through National Biodiversity Strategies and Action Plans (NBSAPs) and national reports, but it falls short in translating global targets into national policy, as evidenced by the failure to meet any of the Aichi Biodiversity Targets from the last decade. Similarly, the effectiveness of the climate NDCs under the Paris Agreement is still under scrutiny. Recent

literature highlights the need for improved compliance mechanisms and greater state accountability to address these challenges. For instance, Koh et al. (2022) discuss how insights from international human rights mechanisms can enhance compliance with the CBD by bridging the gap between reporting and implementation. They argue that integrating accountability measures from human rights frameworks can provide valuable lessons for strengthening environmental agreements.

– Sustainable Trade Policies: These policies can include measures such as enforcing stricter environmental standards for imported and exported goods, which can be achieved through environmental certification schemes like the Forest Stewardship Council (FSC) for timber products or the Marine Stewardship Council (MSC) for seafood. Promoting fair trade practices, such as those certified by Fair Trade International, ensures that producers in developing countries receive fair compensation and work under environmentally sustainable conditions. An example of a comprehensive policy is



the EU Deforestation Regulation (EUDR), which aims to reduce illegal deforestation by ensuring that products sold in
the EU are deforestation-free, replacing the previous EU Timber Regulation that focused only on timber. This regulation
applies to a broader range of products, including soy, palm oil, and coffee, to combat deforestation globally. Additionally,
providing incentives for businesses to adopt sustainable supply chains can involve tax breaks, subsidies, or grants for
companies that implement green practices, such as reducing emissions, conserving water, and minimizing waste (OECD,
590      2020).

– Transnational Conservation Initiatives: Organizations like the Amazon Cooperation Treaty Organization (ACTO) facil-
itate collaboration between countries sharing critical ecosystems to implement joint conservation strategies and combat
illegal activities like deforestation and wildlife trafficking (Fernandes et al., 2024). For example, ACTO member coun-
tries work together on projects that monitor deforestation rates using satellite technology, restore degraded lands, and
promote sustainable livelihoods for local communities. Another initiative is the Great Green Wall project in Africa, which
spans over 20 countries and aims to combat desertification, restore degraded landscapes, and improve food security by
creating a mosaic of green and productive landscapes across the Sahel region (UNCCD, 2016).

These policy packages not only help to mitigate the environmental impacts of global trade but also foster a spirit of inter-
national cooperation, ensuring that environmental protection efforts are harmonized and effective across borders. By aligning
national policies with international standards and collaborating on shared goals, countries can collectively reduce resource
extraction, minimize environmental degradation, and enhance the resilience of ecosystems globally.

### 3.6.4   Recommendations:

– To enhance the effectiveness of international environmental agreements, robust compliance mechanisms must be im-
plemented, using independent monitoring bodies and digital tools for accountability. Regular reviews and transparent
reporting hold states accountable with clear benchmarks and timelines, including mandatory progress reports and public
disclosure. Cross-sectoral integration aligns biodiversity and climate goals with national development plans, creating
synergies across agriculture, energy, and infrastructure.

– Providing technical and financial support to developing countries empowers them with training, data infrastructure, and
resources for monitoring. Engaging stakeholders like civil society, indigenous communities, and private sector actors
ensures diverse perspectives and innovative solutions, fostering grassroots support and inclusive decision-making.

– Mandating environmental certification for imports and exports is crucial to ensure products meet high environmental
standards, which can be achieved by expanding existing certification schemes and developing new ones tailored to
different sectors. Promoting fair trade practices ensures equitable compensation and sustainable working conditions for
producers in developing countries, supported by market access and capacity-building programs. Encouraging regions to
adopt regulations similar to the EU Deforestation Regulation (EUDR) can help ensure products are deforestation-free
and support biodiversity conservation by expanding regulation scopes and compliance verification





– Offering incentives like tax breaks, subsidies, or grants to businesses that adopt sustainable practices can reduce emissions, water use, and waste production. Collaborating with international trade organizations is essential to harmonize standards and promote sustainable trade policies that balance environmental protection with economic development

– Enhancing transnational cooperation through initiatives like the Amazon Cooperation Treaty Organization (ACTO) can lead to more effective conservation strategies by combating illegal activities such as deforestation and wildlife trafficking. Large-scale restoration projects, such as the Great Green Wall in Africa, address desertification, create jobs, and improve food security. Utilizing satellite and remote sensing technologies enables real-time monitoring of environmental changes, informing timely conservation efforts and policy decisions. Supporting sustainable livelihoods for local
communities integrates economic opportunities with conservation goals, reducing reliance on unsustainable practices. Multilateral partnerships between governments, NGOs, and international organizations align conservation with sustainable development goals, ensuring comprehensive outcomes for global ecosystems.

### 3.7 Convivial conservation offers as a set of governance principles for the future of conservation efforts

#### 3.7.1 Background

In a world where biodiversity continues to deteriorate at an alarming rate, the need for innovative conservation and restoration strategies has never been more urgent. By integrating a diversity of knowledge systems and considering relational values when planning relational values into conservation efforts, we can develop more holistic and sustainable approaches to safeguarding biodiversity, ultimately ensuring the health and prosperity of both nature and humanity (see also Section 3.5, 3.6).

Convivial conservation is a new "vision, a politics and a set of governance principles for the future of conservation" (Büscher
and Fletcher, 2019, p.284). Through its core focus on 'living with' biodiversity within planetary boundaries, it closely aligns with transformative action for climate change (Pörtner et al., 2021b). Grounded in political ecology it foregrounds political economy as a significant constraint to transformative conservation. Political ecology is inherently cross-scalar, charting connections from the global to the local, while emphasizing the importance of history and power relations (Watts, 2017). Based on this perspective and allied with social and environmental movements (e.g. Indigenous and decolonial), it proposes "a post-
capitalist approach to conservation that promotes radical equity, structural transformation and environmental justice and so contributes to an overarching movement to create a more equal and sustainable world" (Büscher and Fletcher, 2019, p.283).

#### 3.7.2 Challenges

Convivial conservation responds to two dominating conservation agendas which are presented here as 'strong' versions to help differentiate the contribution it is making. First, so-called 'new conservation', which breaks with a long-standing fixation on
'pristine wilderness' seen as separate from humans, and instead promotes integrated "rambunctious gardens" (Kareiva et al., 2011; Marris, 2013) as cultural land and seascapes. New conservationists propose nature should be integrated into human development (Sullivan, 2006; Buscher and Fletcher, 2020) but do not address the harmful capitalist model of economic development that underpins biodiversity loss (e.g. tourism or Payments for Ecosystem Services). The second approach, 'neo-protectionism',





tries to separate nature entirely from human development, calling for a massive expansion of conventional 'fortress' style pro-
tected areas and so reinforces nature-culture dichotomies (Hutton et al., 2005; Wuerthner et al., 2015; Buscher and Fletcher,
2020). While new conservation moves beyond these dualisms it looks to market mechanisms to fund and save nature (e.g.
Payments for Ecosystem Services, ecotourism) thereby producing other contradictions. Convivial conservation proposes that
both approaches have limited use, as inherited from philosophies and global development models which drive the intertwined
biodiversity and climate crises.

### 3.7.3   Offering solutions

The specific contribution of convivial conservation is that it aims to produce integrated nature-culture spaces within post-
capitalist conservation strategies. At its core it investigates and challenges dominant global political-economic structures,
assumptions, beliefs and knowledge production systems, "including those that are the foundation of paradigms of economic
growth and adaptation without limits" (O'Brien and Barnett, 2013, p.385).

Convivial conservation is gaining traction in research, policy and practice (Massarella et al., 2023; Ochieng et al., 2023).
Today, "there is widespread agreement that our current reality of global, human-induced ecosystemic and climatic change
presents stark challenges for conservation. It is concern for this dynamic that has led to the radical proposals now on the
table" (Büscher and Fletcher, 2019, p.285). At the same time, breaking through the hegemony of protectionist, neoliberal
conservation (Fletcher, 2023) is also convivial conservation's biggest challenge. To further address this challenge, a manifesto
was developed that outlines 10 principles core to convivial conservation. We summarize key elements of these principles here;
for a full overview of all 10 principles we refer to the manifesto website (Conservation, 2024).

1. Integrated landscapes: Humans have always shaped the ecosystems in which they live, co-producing diverse landscapes
   that in turn shaped and supported people. Yet mainstream conservation interventions often separate people from the sur-
   rounding ecosystem based on the unfounded assumption that local communities threaten biodiversity Brockington et al.
(2012). This assumption is undermined by growing evidence that humans, especially Indigenous peoples, have actively
   managed and used what are today erroneously considered 'wild' areas throughout the world (Merino and Gustafsson,
   2021). There is a need to promote landscapes that integrate people and nonhuman species: the question, going forward,
   is not whether people should live with the rest of nature, but how we do (see Section 3.5 and 3.9).

2. Direct democratic and equitable governance: International and regional inequality contributes to the destruction of the
global commons necessitating equitable stewardship of ecosystems, centered around those who live within them. Nur-
   turing extra-local commons institutions and economies based on values of responsibility and care would help cross-
   generational and cross-scale conviviality. Convivial conservation challenges dominant top-down forms of political power,
   advocating for inclusive deliberation and decision-making processes in particular for those in proximity and dependent
   on the ecosystems in question (Lanjouw, 2021). This is based on the principle of subsidiarity, which means that all de-
cisions that can effectively be reached at a local level, should be, with higher-level processes supporting local autonomy
   and only intervening when necessary (e.g., Gokkon, 2018, see also Section 3.6).





3. Non-market, redistributive funding and valuation based on intrinsic/spiritual significance: Emphasizing the monetary valuation of biodiversity is dangerous and counterproductive . Instruments like 'payments for environmental services', REDD+, and carbon credits employ the logic of the problem (capitalist accumulation through natural resource use) as the logic of the solution (Fletcher, 2023). Monetary valuation also conflicts with convivial coexistence between humans and nonhumans, and undermines other non-monetary ways of valuing nature. Delinking conservation from global capitalism could support traditional livelihoods, rather than coercing local people into 'alternative livelihoods' dependent on unreliable and exploitative external markets. Moreover, mechanisms to redistribute existing wealth and resources would help support new livelihoods while precluding the need to fund conservation through environmentally harmful economic growth (Moranta et al., 2022).

4. Embracing diverse forms of knowing: Protected areas have usually depended on Western scientific knowledge paradigms at the expense of rich local and Indigenous philosophies, histories and practices. Yet many diverse other ways of knowing and practical ways of being in relation with the world like Ubuntu (Mabele et al., 2022), Buen Vivir, and Eco-Swaraj promote life through mutual caring and sharing between and among humans and nonhumans, discouraging individualism and unsustainable extraction (Dickson-Hoyle et al., 2022). Local knowledge held by stewards of landscapes and place-based communities are also invaluable and often overlooked in technocratic decision making (Gielen et al., 2024). This full range of diverse knowledge must be valued through respectful partnerships rather than tokenism or extractivism (Orlove et al., 2023).

5. Challenging broader political-economic forces: While Indigenous Peoples and local communities should be supported and have their rights recognized, they should not be made solely responsible for conserving nature. Too often, those living in or close to conservation areas are expected to change their behavior the most (Brockington et al., 2012; Merino and Gustafsson, 2021). But large industrial extractive practices and the elites' high consumerism drive disproportionate biodiversity loss. Yet these people and organizations are not perceived as such because they are far from conservation spaces and appear too powerful and intractable to influence (Wiedmann et al., 2020). Conservationists must avoid appeasing and overlooking the impacts of these forces, and instead challenge both the regimes that indulge in human rights violations and displacement in the name of biodiversity, and the rights of global or national elites to control or hinder conservation efforts (see also Section 3.6.

### 3.7.4 Recommendations:

– promote landscapes that integrate people and nonhuman species

– decisions that can effectively be reached at a local level, should be, with higher-level processes supporting local autonomy

– mechanisms to redistribute existing wealth and resources would help support new livelihoods while precluding the need to fund conservation through environmentally harmful economic growth

– The full range of diverse knowledge must be valued through respectful partnerships





### 3.8  The social-economic value of ecosystems will increase in proportion to rising real market incomes and the changing scarcities of ecosystems

#### 3.8.1  Background

People derive various benefits from nature, such as through biodiversity, ecosystems or ecosystem functioning. These benefits can manifest as tangible outputs, such as water and food, but also include cultural, recreational, and spiritual interactions that directly or indirectly influence human well-being (e.g., Pascual et al., 2023). One way to conceptualize these benefits is through the notion of ecosystem services that include both use and non-use values of nature. The values in this category are anthropocentric, encompassing both instrumental and relational values (IPBES, 2019). The continuous loss of animal and plant species and their respective habitats leads to the loss of the services they provide. To be better able to reflect these ecosystem services in benefit-cost analyses, environmental-economic national accounting or damage litigation processes, governments convert ecosystem services into monetary values (Bishop et al., 2017). Although assigning monetary values to ecosystem services involves numerous philosophical and practical challenges, the alternative is often to consider no value at all, leading to an underinvestment in ecosystems (Dasgupta and Treasury, 2022). Thus, already in 2010, at the 10th Conference of the Convention on Biological Diversity in Japan, the international community agreed that the values of biodiversity needed to be integrated into planning processes (Aichi Target 2). In the Kunming-Montreal Global Biodiversity Framework it is reflected in Target 14: Integrate Biodiversity in Decision-Making at Every Level.

#### 3.8.2  Challenges

Governments around the world are currently looking for new approaches to appropriately assess the benefits from scarce ecosystems and their economic value. This is intended to assist in making the consequences of the destruction or the benefits of the conservation of nature more visible in analyses that underpin political decision-making processes and help with an economically efficient and environmentally effective allocation of tight governmental budgets.

For now, calculation methods of nature's values incorporate—if at all—solely the monetary value of ecosystem services as determined under current conditions (Drupp et al., 2024), meaning that nature becomes relatively less valuable over time compared to other goods and services whose value increases with the expected rise in global economic prosperity. In fact also our appreciation of nature increases over time as we get wealthier and ecosystems scarcer. Two factors play a key role in this changing value of scarce ecosystems over time. The prosperity of the world's population is expected to rise—by an estimated inflation-adjusted two percent per year (Müller et al., 2022)—and as household incomes increase, people will be willing to pay more to conserve nature and enjoy its services in the future. In addition, as the services provided by ecosystems become scarcer, this will further increase their value to society. The fact that scarce goods become more expensive is a fundamental principle in economics, and it also applies to nature's values.





### 3.8.3 Offering solutions

Drupp et al. (2024) provide governments with a ready to use formula to estimate the future economic values of scarce ecosystem services that can be used in decision-making processes. The formula scrutinizes up-to-date evidence on the so-called relative price change of non-market environmental goods (e.g., Hoel and Sterner, 2007; Sterner and Persson, 2008; Drupp and Hänsel, 2021) and recommends considering nature's values to increase proportionally with real market income. This is in line with what governmental bodies use for valuing reductions in mortality risk or travel time. As a result, if only the expected increases

in income over the next 100 years were taken into account, the value of global ecosystems would have to increase by more than 130%. This holds for stagnating ecosystems. If ecosystems are projected to decline or degrade further, the value adjustment needs to be higher still. In the case of endangered species as captured in the prominent Red List Index, for instance, the value adjustment would amount to more than 180%. Accounting for these effects would thus increase the likelihood of projects that conserve ecosystem services to pass a benefit-cost test.

Drupp and Hänsel (2021) apply the formula to the evaluation of global climate policy. Economists typically use integrated climate-economy assessment models, such as the DICE model developed by Nobel Laureate William Nordhaus, to evaluate the trade-offs between mitigation costs and avoided damages from climate change and to estimate required $CO_2$-prices (Nordhaus, 2019). A key criticism leveled at these models is that they do not appropriately capture the loss of nature's services and thus underestimate climate damages. Drupp and Hänsel (2021) disentangle how non-market goods and services, such as

environmental amenities, are captured within these models and explicitly account for this based on an empirical analysis of fundamental drivers of the relative price effect of non-market goods. They find that the social costs of climate change increase by more than 50%, suggesting substantially higher economically optimal $CO_2$-prices (see also Section 3.6 and 3.9). The increase in the economically optimal global mean temperature change is accordingly reduced by half a degree Celsius, which highlights the importance of accounting for the scarcity of nature when evaluating climate policy.

### 765 3.8.4 Recommendations:

- Future benefits derived from ecosystems should be uplifted proportionally with increasing real market incomes and changing real scarcities of ecosystems.

- Estimates for the aggregate societal cost of emitting an additional ton of carbon into the atmosphere, i.e. the social cost of carbon, and optimal carbon prices should be adjusted to reflect how climate change affects the changing scarcity of
ecosystem services.





## 3.9 Share and strengthen frameworks for biodiversity governance of revitalized, just human-nature relationships and future biodiversity conservation

### 3.9.1 Background

Todays' Earth dominating production and consumption patterns are far from achieving the CBD 2050 vision of 'living in
harmony with nature', even those under climate scenarios that are currently considered as 'most sustainable' (SSP1, RCP 2.6
- Pereira et al., 2020b, 2024). While the world may focus on achieving climate targets, we are undermining our shared life-
support system and missing out on plenty of synergies revitalizing human-nature relationships and at the same time mitigating
climate impacts. Therefore, national and local governments need to step up their efforts to effectively prioritize the Kunming-
Montreal Global Biodiversity Framework (GBF) in order to halt and reverse worsening biodiversity trends in an explicit
biodiversity governance effort (Obura et al., 2023; Kim et al., 2023).

### 3.9.2 Challenges

The major drivers of biodiversity loss are rooted in production and consumption patterns, which result in land and sea use
change, resource overexploitation, pollution, and climate change (IPBES, 2019). Thereby industrial agriculture practices were
observed to be contributing more to global biodiversity loss than direct climate impacts (Sánchez-Bayo and Wyckhuys, 2019;
Jaureguiberry et al., 2022, see also Section 3.4, 3.5, 3.6). Other drivers to biodiversity loss are human population growth along-
side growing inequality, harmful subsidies, and particularly those that aggravate the reliance on fossil fuels Kim et al. (2023).
A critical reflection on often unintended repercussions from harmful subsidies on multiple biodiversity values is needed, as for
example performed in Switzerland (Gubler et al., 2020, see also Section 3.7 and 3.8), as well as the larger-scale mechanisms
within the broader economy are under inspection for finally reversing declining biodiversity trends (Otero et al., 2024). Pivotal
agreements on reducing subsidies affecting ocean fisheries, such as fuel subsidies, may serve as examples for a fundamen-
tal change (Sumaila et al., 2024). To systematically address and reverse declining trends in all ecosystems, the relationships
between people and nature need to be revitalized at the same time as collective action and agency over those relationships is
regained (Pickering, 2023). In terms of biodiversity governance, the main challenges include missing platforms i) for reflecting
on norms as well as setting norms on multiple societal levels, ii) for addressing injustices, and iii) for enforcing accountability
in ecosystems damaged by exploitative actions which are often also rooted in colonial histories (ibid.)

### 3.9.3 Offering solutions

The following framework by Perino et al. (2022) promises to improve future action reversing current trends of declining
biodiversity:

- The identification process for locally suitable actions and the promotion of stakeholder ownership must recognise the
multiple values of biodiversity (Pascual et al., 2023; Martin et al., 2024) and account for remote responsibility.



– Cross-sectoral implementation and mainstreaming of biodiversity considerations need scalable and multifunctional approaches to restoring ecosystems and aim for positive futures for nature and people.

– Assessment of progress and adaptive management needs to be informed by novel biodiversity monitoring and modeling approaches that address the multidimensionality of biodiversity change, including the incorporation of Indigenous and local knowledge (as e.g. in Gielen et al., 2024).

In large, biodiversity governance seeks to achieve compliance with CBD objectives, including entrenched principles of equity and application of human rights. Implementing global environmental targets requires an equity lens and a rights-based approach, as projects that are aligned with local people's preferences and through inclusive governance are likely to have more effective social and ecological outcomes (Obura et al., 2023; Löfqvist et al., 2023; McDermott et al., 2023). The GBF explicitly states that the 'implementation of the framework should follow a human rights-based approach respecting, protecting, promoting and fulfilling human rights' (CBD/COP/DEC/15/4). Moreover, unpacking elements of social and environmental justice, including procedural, recognitional and distributive dimensions, is needed to support a long-term transformation towards sustainability (Leach et al., 2018; Pereira et al., 2023). For policy-making, an integration is needed alongside the Sustainable Development Goals where synergies and trade-offs are mapped with the goals on biodiversity Pörtner et al. (2021b). In addition to biodiversity monitoring, accountability, and resource mobilization, the (in-)justices that are implied in decision-making processes need to be reflected with marginalized groups shaping the collective process and outcome of biodiversity as generative, life-supporting value. Indigenous Peoples and local communities lead by example by stewarding the biosphere supporting ecological integrity and thus biodiversity conservation (Garnett et al., 2018; Dawson et al., 2024; Seebens et al., 2024; Massarella et al., 2021, see also section ), see also section 3.6 and 3.7).

The nature futures framework (NFF) supports the facilitation of future making processes where multiple values of nature are recognized, deliberated about by diverse interest groups, and actions are chosen based on the collectively shared nature values (Pereira et al., 2020a; Kim et al., 2023; IPBES, 2023). The NFF invites adaptive and anticipatory management which can be supported by recurrent scenario analyses to explore and plan for pathways to positive futures that not only mitigate negative outcomes but also enhance positive synergies between biodiversity conservation and climate action. Immediate actions include:

– integrating plural values and engaging diverse stakeholders in decision-making processes,

– mainstreaming biodiversity conservation into all sectors,

– use nexus approaches to address interlinkages, co-benefits, and trade-offs between sectors,

– improving policy coherence and integration, and

– applying known best practices in ecosystem restoration and management (see also Pörtner et al., 2021a, Section 7).

By leveraging these strategies, we can align biodiversity conservation with climate mitigation efforts, fostering a holistic approach to secure a sustainable future for both people and nature.



### 3.9.4 Recommendations:

– National and sub-national policies should focus on broad-based social safety nets to initiate and reinforce conservation of biodiversity in revitalized human-nature relationships.

– Encourage biodiversity governance institutions to allow for norm reflection concerning ecosystems and facilitating a just decision-making process in compliance with CBD objectives and human rights.

– While prioritizing 1. alongside biodiversity conservation measures, mobilize resources through multilateral benefit-sharing mechanisms (Nagoya Protocol, Article 10)

– Share and extend existing guidance for assessing equity in protected area management as a standard, to hold decision-
makers accountable and apply to other areas of biodiversity governance.

## 4 Conclusions

The nine themes presented here highlight the complex interrelationships within the biosphere and their connections to social and economic systems, and how these are entangled with climate change. We are trapped in various vicious cycle. For example, changes in temperature and precipitation patterns as a result of climate change and deforestation can lead to lower agricultural
yields and increased fires. This increases pressure on ecosystems and local, human communities, which depend on them, and which are under pressure to provide resources and products for the global market. The provision of various commodities under current trading paradigms and subsidy schemes further fuels climate change, ecosystem degradation and deforestation. In addition to identifying interdependence between these challenges, our nine themes offer some insight into escape hatches from this vicious cycle. Our major insights are:

As **climate change** alters weather patterns, disrupts hydrological cycles, and increases the frequency of extreme events, it undermines ecosystems' capacity to provide essential services such as water purification, pollination, and soil fertility (IPBES, 2019). To mitigate these impacts, it is crucial to stop greenhouse gas emissions **and** stay close to, or even below, the 1.5°C target. Exceeding this threshold, for example, is expected to cause widespread mangrove retreat (Section 3.1) and increase the severity of fire weather across most of the world (Section 3.3). Regions that are not adapted to fire regimes may experience
increased vulnerability, with adaptation efforts driving up costs (Section 3.3). Beyond stopping greenhouse gas emissions, it is essential to reduce **anthropogenic stressors** such as overexploitation, resource extraction, pollution, and unsustainable management, which threaten the ability of ecosystems to directly or indirectly support human well-being, economic stability, and overall quality of life (Sections 3.1, 3.2, 3.4, 3.5,3.6).

**Nature-based solutions** to climate change have demonstrated **co-benefits** for biodiversity and hydrological cycles (Section
3.1, 3.2). However, afforestation, for example, requires careful planning and management to avoid unintended ecological consequences (Section 3.2, Section 3.3). Controlled burning is a natural fire management solution that can maintain fire-dependent vegetation types and biodiversity and is cost-effective, compared to other engineered solutions (Section 3.3). Quantifying the



co-benefits of dynamic ecosystems over time and across spatial scales for decision making remains a challenge. One approach to accounting for dynamic carbon storage is to measure it in tonne-years, which are proportional to degree-years of avoided

warming (Section 3.4). To maintain and/or restore various **ecosystem services**, at least 20-25% per square kilometer of (semi-)natural habitats should be present in human-modified landscapes, including agricultural and urban areas (Section 3.5). Given the benefits of nature-based solutions that go beyond climate change mitigation, future efforts need to focus on improving metrics to monitor and quantify progress as well as analysing and sharing successful management practices (Section 3.2, 3.3, 3.5, 3.6, 3.9).

**Global trade** is a major contributor to ecosystem vulnerability, degradation and social inequalities in regions far from where products are consumed, resulting in uneven environmental impacts and exacerbating socio-economic disparities (Section 3.6). To address these issues, it is essential to adopt circular economy principles and **international** comprehensive **policy frameworks** that recognise the multiple benefits of nature-based solutions and integrate climate action with economic and social objectives (Sections 3.6, 3.8, 3.9). For example, the value of ecosystem services should be adjusted in proportion to

rising real market incomes and changing ecosystem scarcities (Section 3.8). Such policy frameworks and economic measures can create synergies that enhance ecosystem resilience, support sustainable development and improve the health of human-nature relationships (Sections 3.6, 3.7, 3.9).

While international cooperation is essential to coordinate policy frameworks, **local implementation** strategies need to be tailored to local circumstances and conditions, taking into account all available knowledge (Section 3.7). For example, man-

made changes in species composition (through resource management or introduction of new species) needs to take into account changes in biogeochemical cycles and impacts on local biodiversity 3.1, 3.2, 3.3). The development of adaptation strategies to fire, especially in previously unaffected regions, is crucial as fires can lead to significant carbon losses and the assessment of future fire risks remains uncertain (Section 3.3). It is important to explicitly consider temporal dynamics and spatial heterogeneity in decision-making processes (Sections 3.4 and 3.5, 3.7). Indigenous peoples and local communities are often an example

of effective stewardship of the biosphere and support ecological integrity and biodiversity conservation (Section 3.9. Implementation decisions should therefore be taken at the local level where possible, with higher level processes supporting local autonomy and governance (Sections 3.6, 3.7, 3.9). However, governance from national to local scales tends to be influenced by the geographical context, resource availability, time horizon and socio-political dimensions (e.g., Nilsson et al., 2018). Therefore, more local studies and effective knowledge sharing, together with advances in the ecological theory of transient

ecosystems, will help to develop future proactive solutions.

Taken together, the nine themes illustrate the importance of considering the impact of human activities on neighbouring areas when analysing, evaluating or developing policies or economic measures. Focusing exclusively on a single problem, question or objective is not enought. Overly siloed approaches can overlook, or worse, exacerbate, existing problems in other areas (Fanning et al., 2022). As different aspects of the Earth system crisis are typically addressed by different research disciplines,

closer collaboration between scientists of different research disciplines is essential to develop a holistic understanding and to effectively address complex, critical issues even in smaller research projects. This is already practiced in the major reports of e.g. IPBES and IPCC.



Our study offers a sample of the active, growing body of work on biosphere research from the perspective of different research domains, provides recommendations and reasserts the value of interdisciplinary research. Nevertheless, this collection 900 of pressing ecological issues does not claim to be exhaustive, and the compilation may be superficial on some topics that require more in-depth discussion. In the future, we need to encourage greater contributions from scientists in other regions of the world, particularly the Global South, to incorporate their knowledge and perspectives. Their input can help identify new targets and research questions that may have been overlooked so far.

Despite receiving less public attention than other currently dominant issues, the Earth System crisis - including climate 905 change, biodiversity loss, pollution and land-use change - remains the major challenge of this century. While many ecosystems around the world are suffering from these threats, functioning ecosystems also offer significant potential for addressing many aspects of Earth system crises.

In conclusion, we curated this list of pressing environmental issues and recommendations to underscore that we are not limited by how much we know about the problem or how much we know about how to act. The obstacles are structural, 910 cultural and political in nature. They prevent the necessary pace and scale of implementation needed to achieve the various international commitments and the goals of the Paris Agreement and the Kunming-Montreal Framework in a fair and equitable manner. Instead, an effective implementation can promote a flourishing biosphere, that facilitates economic, cultural, and spiritual interactions essential to human well-being

*Author contributions.* Conceiving and designing the study and providing editorial oversight: FJB, NK, AB, AR, RM, GBS. Constituted the 915 Editorial Board to select the themes: AB, LP, GH, CK, AR, AG. Led and coordinated the overall writing: FJB. Lead-authors of writing teams: FJB, GBS, AR, JH, DM, AG, AM, SK, MH, RM. Performed literature review: all. Contributed to the writing: all.

*Competing interests.* one of the co-authors is a member of the editorial board of BG (Anja Rammig)



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
