# Peer review of "Reviews and syntheses: Current perspectives on biosphere research - 2024"

_EGUsphere, 2024_

## Author Response (AR1)

**Referee #1**

In the manuscript *Reviews and syntheses: Current perspectives on biosphere research--2024*, the authors propose that, given the speed at which climate reports are generated and lack of interdisciplinary efforts, the scientific and policy communities need a method to more frequently synthesize the latest biosphere research across disciplines. The authors employ an example survey method and present a review of their findings, which include a set of nine emergent and pressing challenges in ecological research. The goal of this work is to synthesize the latest policy-relevant research and inform interdisciplinary collaborations to arrive at creative, whole-picture solutions to ecological challenges.

Thank you for your constructive and supportive review. Your feedback is invaluable in helping us to communicate our key messages more effectively and to achieve a clearer, more balanced synthesis of the content.

This is a useful concept and one that I think could more rapidly advance Earth systems research, as well as improve our tools for addressing climate change.  I do wonder if the emphasis on including almost exclusively recent works might exclude issues that have accumulated a large body of work over time but that still have advances in recent years, ultimately reducing some continuity in addressing climate challenges. For example, I was surprised that some topics, like changing carbon cycle dynamics or fossil fuel consumption, weren't given a section of the report, although the authors note their importance as a top priority for mitigating climate change in the introduction. It's possible those topics weren't singled out because they were either meant to be woven throughout the sections, or that this paper is meant to be complementary to other documents like IPCC reports or the Global Carbon Budget where that is the emphasis. Either way, I think the authors could make this intent clearer up front.

We aimed to focus with this paper on recent insights related to the biosphere. Hence, drastically reducing fossil fuel consumption - essential for fighting climate change - is not addressed  in this paper. We made it clearer and will rewrite the introduction. Furthermore,  we now mention long lasting insights documented in IPCC and IPBES and other reports in all background or challenge sections.

 I ultimately wished for more of a conclusive statement of implications in both the introduction and conclusion detailing the novelty of the synthesis and its suggested action steps, as well as the role of this sort of survey in making ecological advances. Including this text might also help highlight broader issues that were not specific sections of the article but that are critical steps for addressing ecological challenges.

 With this study, we aim to raise awareness of the various challenges within the biosphere and their interconnectedness with other crises within the Earth system, provide synergistic strategies to address complex challenges, and stimulate future research questions.

We revised the text to better reflect the purpose of this manuscript in the introduction
We revised the Synthesis and Conclusion to include findings on integrated decision making,

land management strategies, measurement and knowledge generation, and linked them to species and climate change objectives.

It does seem like there is a slightly different authorial voice in some of the sections, as though different sections were written by different people, which makes some of the sections fit together differently than others (for example some sections include a statement of the problem/question in the 'Background' subsection, while others do not and include the problem only in the 'Challenges' subsection).

Some sections are also much longer, and although I don't think any of the sections are meant to be emphasized more than another, it does have the effect of giving the policy sections a bit more weight. I think it might be helpful to trim the sections a little so that they are more consistent formats and lengths. This might also improve some of the repetitiveness that developed in the later sections of the article.

We made the following revisions to harmonise the topic sections:

- Each section has been restructured to a maximum of 1500 words for consistency.
- The *background* sections have been revised to create a seamless transition from the *introduction* without redundancy.
- We improved the *challenge* sections with clear problem descriptions and ensured coherence with the solutions presented later.
- General recommendations have been avoided in the *solution sections*.
- For certain topics, specifically 3.4, 3.6, 3.7, 3.8, we now provide a more comprehensive overview of each topic, including additional references.
- Finally, a core team has reviewed and polished the text to ensure consistency and cohesion.

The last 4 sections especially (3.6-3.9) seem to have a lot of overlap and I wonder if there could be different ways to merge some of these ideas that could also reduce the length. Maybe these last sections could be reorganized into a section on biodiversity and conservation, a section on social ecological challenges and inclusivity, and a section on directions for science policy and economics. Alternately, sections 3.7 and 3.8 seem to pose oppositional ideas about including ecosystems in social-economic frameworks and I wonder if merging these sections to highlight the controversy growing in this area might be a way forward. Sections 3.6 and 3.9 also feel quite similar so may be able to be merged.

I do think it's important to highlight that these themes are interconnected as the authors have done, which ultimately means some ideas may be reiterated, but I think there may be ways to trim the text that maintains the spirit of that idea while still making the final sections more distinct.

We refined the focus of each section to improve clarity and reduce overlap.

More specifically:

- Section 3.6 focused on holistic policy packages at national and international levels.

- Section 3.7 (now 3.8) focused on the long-term principles of local solutions and inclusive, community-based conservation.
- Section 3.8 (now 3.7) analysed the short-term societal value of ecosystems.
- We have changed the order of 3.7 to 3.8.
- We have integrated section 3.9 with section 3.6.

The aim of this reorganisation is to improve the coherence and thematic focus of the topics while maintaining the links between them.

A final thought for the authors to take or leave--it could be nice to conclude with some sort of conceptual figure that ties all the sections together. Something demonstrating the type of local to global collaboration and action needed across different sectors and disciplines to address specific challenges would be nice. I don't think this is absolutely necessary but would be great to make the sections feel a bit more cohesive and give the reader a visual overview about how all these parts fit together.

We worked on a graphic, but eventually discarded it as it did not achieve our aim of making the content much more understandable and accessible.

Overall I do think this is a nice summary of many emerging biosphere topics, and the methodology used by the authors, especially if they are able to continue expanding the reach of their solicitation, could be a useful way to identify key areas where research and policy advances are needed.
Thank you very much for these motivating final words.

Section and line specific comments are included below.

We have revised the text based on your helpful and constructive comments. Thank you for your support.

**Abstract:**

Line 3: At the start of the abstract it would be great to add a little detail about the methodology since it seems like that is an important part of what the authors are trying to test as a strategy for generating a report.

It would be great if the authors could include an overarching sentence at the end of the abstract stating why the review is important.

We revised the abstract following your suggestions.

**Intro:**

 Line 2: Changes should be changed. First sentence is a little awkward.
We reformulated the sentence.

Line 20: natural processes

corrected

Paragraph starting line 45: This seems like information that should have been included in the previous paragraphs when the IPCC reports are introduced. I see how the authors are trying to use it to lead into the idea that these reports are not interdisciplinarily integrative, but if the paragraph stays here, the authors need to make that point more explicit in this paragraph.

We have substantially revised the section to discuss (i) interdisciplinary but not regularly updated special reports of the IPCC and IPBES, (ii) topic-specific, regularly repeated reports that sometimes lack an interdisciplinary background, and (iii) the 10NICS report, which is exemplary for this paper and focuses on insights in the context of climate change.

Line 53: The authors note here that there is a lack of integrative reporting on the biosphere but I'm not sure the previously stated information convinces me of that entirely, as written. It may be the case for the reports included in the previous paragraph, but could the authors give some explicit examples there?
We have narrowed the statement and linked this paragraph more closely to the previous paragraph to assert that there is no report comparable to 10NICS, but that it focuses on the biosphere, is organised interdisciplinarily and is published annually.

The final paragraph of the introduction feels a little more like a conclusion. I wonder if this paragraph could be removed or changed to refocus on the idea started in line 75 about climate mitigation and carbon emissions to detail how this overarching concept is woven throughout the 9 key topics.

We follow with this paragraph the structure of the 10NICS-Reports which close the introduction with a short summary of the topics discussed on the following pages.

**Methods:**

It would be nice to have a little more specific criteria about how these 9 proposal topics were judged and selected aside number of citations of recent literature. Was there a rubric for selection?
The three criteria were (i) sufficient evidence from peer-reviewed publications in the last two years; (ii) no ongoing critical debate on the issue; (iii) relevance to international negotiations, which are very similar to those used for 10NICS. These criteria were also communicated in the questionnaire. We have also attached an appendix to the questionnaire we used. We have chosen the appendix to provide this information because we want this manuscript to focus on the latest findings and only to a limited extent on the methodology.
Only authors were selected according to their number of citations in recent literature. We rephrase the sentence.

**Insights:**

**3.1 Coastal habitats**

Line 127: Breaking up this sentence would make the ideas presented clearer.

done

Lines 126-137: This paragraph seems more solutions oriented rather than a description of challenges. Possibly move to the next section?

done - modified to a problem description instead of moving to another section

Line 157-159: This short paragraph seems a little out of place. Maybe it could be merged with the previous paragraph?

Done - modified to improve clarity. Some part must have gotten lost due to heavy editing

**3.2 Forests and precipitation**

The background highlights the important water cycling functions of forests but doesn't state the question/problem. This might be okay, but other sections do state the problem in the background so it would be nice to make them more consistent.

The problem is consistently stated in the challenge section in all topics.

Lines 217-219: The authors note here that evidence is mounting so it seems like it might be important to have multiple references to support the claim, especially since the next sentence notes that, despite this evidence, the effect is not observed in the tropics.

We reformulated the sentence.

Line 221—I don't follow this sentence "This suggests that …elsewhere."

We removed this sentence.

Line 230: "Therefore, also…" is a little odd grammatically. Maybe take out the word "also"?

We reformulated the sentence.

Line 241: Extra parentheses are included here that aren't needed.

deleted.

Line 245: The wording and use of colon here is a little awkward. I think I would switch to "Reforestation has larger effects on several areas of the globe" and leave out the colon (or something along those lines).

We reformulated the sentence, following your suggestion - Thank you.

Line 235/246:  I follow the use of the tipping point term that the authors use in both locations, but the description is somewhat vague. It would help to be explicit what you mean

here—maybe "a tipping point at which a forest transitions to a dryland or grassland due to decreased moisture"?

We reformulated both sentences, again following your suggestion - Thank you.

**3.3 Fire risks**

Line 284: Parenthesis needed before Copernicus rather than after. done

**3.5 20-25% natural habitat in modified landscapes**

This section ultimately posits a divide between humans and nature. The following sections discuss the inclusion of humans in nature and recognizing the intrinsic value of the natural world. It might be useful to have some text that discusses these different ways of viewing nature and their relevance to how we make scientific and policy decisions.

**Response**:Thank you for your comment.The intent of the section is not to isolate humans from nature but rather to foster mutually beneficial interactions within integrated landscapes. We have added some nuances for clarification.

*Line XX: The 25% high-functioning nature in every square kilometer offers a key policy tool since it is the first widely applicable measurement of the minimum level of human-modified land that needs to be in a (semi-)natural state across several NCP and a wide range of landscapes. This proposed habitat level is the minimum level, not the optimal level required to meet adequate NCP demand \citep{mohamed_securing_2024}. This habitat threshold reflects an approach that harmonizes human activities with ecosystem integrity, focusing on integration rather than strict separation between human and nature. It serves as a general guideline synergizing with existing policy targets (e.g., UN Decade on Restoration) for prioritizing conservation initiatives and formulating adaptive, scalable policies beyond traditional natural areas.*

Line 472: Is there any indication that the spatial arrangement matters? For example, does each individual square kilometer need to be 20-25% natural habitat (so habitat patches) or if you have 2 square kilometers of land but the natural habitat for both is all shifted all to one side in a continuous tract does that achieve the same result? I'm not sure whether there's an answer but just curious.

**Response**:The spatial arrangement of natural habitats indeed depends on the specific ecological functions or NCP targeted. Studies indicate that having natural habitats distributed into dispersed patches across each square kilometer support more consistent NCP delivery, particularly for NCP that rely on proximity, such as pollination, pest control.or recreation (Garibaldi et al. 2023, 2011, 2021, Fahrig. 2013; Mitchell, et al.2015;Browning et al., 2024). However, for certain functions that do not require such proximity, such as carbon sequestration, contiguous habitat patches could be also beneficial.

 https://doi.org/10.1111/1365-2664.14517

https://doi.org/10.1111/j.1461-0248.2011.01669.x

https://doi.org/10.1111/conl.12773

https://doi.org/10.1111/jbi.12130

https://doi.org/10.1016/j.tree.2015.01.011

https://doi.org/10.1016/j.scitotenv.2023.167739

Line 490: Missing a parenthesis

**Response**: Done

**3.6 Comprehensive policy packages**

The background to this section is much more extensive than the others and states many of the problems upfront while other sections are not organized this way. It would be great to be more consistent between sections.

Line 538-543: This part feels like it belongs in the 'Offering solutions' section.

This section has a lot more detail than the others and it feels like there is more focus here. It might be helpful to trim or to move to the end and merge with section 3.9, which seems quite similar.

**3.7 Convivial conservation**

This section also reads as if written by a different author.

I wonder if the list of principles outlining convivial conservation could be edited to a summary paragraph that states the main intent, or one sentence per bullet point. This seems a little over-detailed, especially as one of 9 sections.

Yes, we changed accordingly and chose it works best here to use one sentence per bullet point.

**3.8 Increasing social economic value of ecosystems**

Line 725: This section seems to be in direct conflict with the previous section's suggestions that we do not use capitalist frameworks to analyze ecologic value. Could the authors address this conflict more completely? It seems like these differing ideas should be discussed, or that these two sections could be one section that describes the current debate and ways forward with it.

We have tried to address the conflict between this and the previous section by making clear upfront that no monetary valuation often means that a zero value for ecosystems is assumed by decision-makers. In addition, we've reformulated the bullet point on economic valuation in

the previous section in a way that underlines that monetary valuation should be complemented by approaches that emphasize the convivial coexistence between humans and nonhumans.

**3.9 Biodiversity governance**

774: Earth dominating is kind of a strange term. Maybe something else? Could also be removed and the sentence would still work.

Suggest to change to "Our current production and consumption patterns…"

The background text for this section is a little vague. It relies heavily on knowing the details of the Montreal Global Biodiversity Framework. It would be great if the authors could add a bit more detail about what specific steps might prioritize the GBF.

Thanks for this, I have added additional text to outline the objectives of the GBF.

788: Here the authors state "as for example in Switzerland…" but do not go on to describe the example and detail how it demonstrates the concept described here. This example should be outlined in more detail as should the "larger scale mechanisms" at the end of the sentence.

**Conclusions**

Line 843: cycle needs an s at the end.

Is there a reason why some words from lines 850-878 are in bold text? If so, could that be made clearer?

We rewrote the conclusion and put the subtopics of the 4 insights in bold to give a quick overview.

Line 850: If the idea here is synthesizing the latest science that has been missed by groups like the IPCC and generating new ideas/solutions, it seems a little odd to me that the primary conclusion is so similar to that which is currently the emphasis of these major organizations. This can be an important point—maybe the authors should emphasize, as proposed by the latest IPCC reports, this 1.5 degrees needs to be our major focus—but presenting it as something new or different seems a bit misleading. It would be great to have a statement of what's new from this synthesis or, alternatively, if this backs up findings from other reports, that should be emphasized.

We rewrote the whole section and synthesise 4 insights based on the previous 8 themes.

Line 868: This seems like something that maybe could be a focus—the need to improve monitoring and quantification metrics. Based on the presented focus areas, how do the authors propose to do so and what specific metrics would they like to see?
Improving the monitoring and sharing of information is one of the 4 insights that we now describe in more detail.

The conclusion paragraphs starting on like 878 are less repetitive/summary type paragraphs and I wonder if the conclusion could be paired down to focus on these rather than spending as much time on summarizing what was written previously.

We hope that the elaboration of 4 findings across the topics will be much more helpful for the readers than the first draft and thus the first part of the conclusion can be kept.
* * *
**Anonymous Referee #2**

This manuscript reviews the literature and provides current perspectives on biosphere research from 2021-2024, using an interdisciplinary perspective. They point to similar reviews of other topics (e.g., top 10 topics in climate change) to justify their approach. They also argue that the slow pace of large-scale international reports (e.g., IPCC's assessment reports) necessitates periodic reviews such as theirs to provide policy relevant information. The goals and objectives of the manuscript are ambitious and laudable, and there are interesting aspects of the synthesis. I feel there is value in the study.

Thank you for your supportive and critical feedback, which lives up to the reputation of 'dreaded reviewer 2' in a helpful way. We appreciate your insights and implement your suggestions in the vast majority of areas, and choose between your alternative suggestions to improve the manuscript, including implementing reviewer 1's suggestions.

Now, if I put on my critical reviewer hat and really embody the dreaded "reviewer 2" … I felt this was very underbaked. I think this manuscript suffers from the "Frankenstein" effect, in that the sections that comprise the synthesis have just been copied and pasted together by the various section leads, without **any polishing or alignment between them**. Some sections are roughly **double the length of others**. There is extensive repetition of information in the latter half of the article. There are also many instances in which boilerplate sentences occur in the latter half of the article, which really do not need to be there. Some of the sections have thoughtful solutions and recommendations that cover a wide base of the challenges, whereas others have extremely cursory recommendations that largely recommend some of the authors' own views. For example, the NCS implementation risk only provides one suggestion – use tonne-year accounting – which one of the authors has published extensively on, without noting the many studies that have pointed out potential issues with this method. There is also no discussion of the many other challenges associated with NCS implementation (additionality, leakage, environmental justice concerns, financing, non-carbon climatic effects, etc.). Many of the sections also present solutions and recommendations that do not relate to the challenges presented. The result is that for many of the sections, the "emerging topics" that are presented feel surface-level and underdeveloped.

I have five major suggestions to improve this manuscript:

1. Identify a subset of the authors (perhaps 1-3) to read through all sections and edit heavily. The paper is very lengthy, and unfortunately today, folks tend to have limited time/energy to read lengthy papers. I would suggest establishing a word limit or some sort of agreement in terms of the maximum length of sections. This subset of

coauthors could read for several things: 1. Is the background of each topic area relevant and necessary, or is it simply repeating elements of the introduction? 2. Do the proposed solutions and recommendations directly address the challenges identified? Similarly, are they specific enough to actually provide novel insight on the topic, or are they overly general (e.g., "develop policy to solve this"). 3. Decide what the purpose of the recommendations section is. Some sections just have cursory bullets, others have thoughtful paragraphs with specific examples. Perhaps these recommendations sections are not needed (see point 3).

We revised the manuscript by undertaking the following steps

- Each section has been restructured to a maximum of 1500 words (including References).
- The *background* sections are revised to create a transition from the *introduction* section without redundancy.
- We improved the *challenge* sections with clear problem descriptions and ensure coherence with the solutions presented later.
- General recommendations have been  avoided in the *solution sections*.
- Finally, a core team reviewed and polished the text to ensure consistency and cohesion.

2. Alternatively, consider rewriting each section to identify one key insight from each topic area. Rather than providing a mini-review of all challenges and trying to highlight multiple solutions and recommendations, pick one insight (ala the Martin et al. 10 insights for climate science piece you reference). I think this would help prevent the reader from feeling like the authors have left out many important areas – as they will expect that you will not address all areas from the beginning. This would also allow you to greatly condense the article and be much more focused. Indeed, the section titles already somewhat suggest this insight – the supporting text just needs to be further developed to more succinctly justify the insight.

We followed your first suggestion and still take some elements from this second point to improve focus and clarity:

- We clearly stated that this list is not exhaustive and that future publications will cover additional topics, as we plan to publish this type of paper on a regular basis (probably annually).
- We streamlined and refined the longer sections, removing repetition to eliminate excessively long passages, so that each topic remains concise and focused.

3. Provide some sort of synthetic thinking across the 9 subject areas. The discussion currently just provides one or two key points from each section, but no synthetic thinking is provided in terms of cross-cutting themes or key areas of research that could address multiple challenges. One suggestion is that rather than providing recommendations (which are typically just bullet points that repeat the information in the solutions section), remove the recommendations and provide a synthetic analysis in or before the discussion… or perhaps compile the recommendations for each topic

in some sort of table. For example, support for increased autonomy of Indigenous and local communities was a common solution – and thus drawing out this theme across the topics could be valuable for your intended audience to see.

Thank you for this valuable suggestion. We revised the discussion section, restructured the synthesis section, and organised our recommendations around the following key findings: improving inclusive decision-making, advancing land management strategies, improving measurement and knowledge generation, and linked these findings to the goals of species conservation and climate change mitigation.

4. Provide more details and information on the actual data collection / whittling of ideas down in supplementary information. As I mention above, I am not convinced that all sections sufficiently address emerging trends across each topic. I know it is impossible to cover everything, but there appears to be a bias towards the work of the coauthors for several of the topics. There should be more transparency in how these topics were arrived at, given that they might influence future policy, research, action, etc.

Thank you for sharing your considerations. We worked on the following points in response to your comment:

- We included an appendix with the questionnaire used in our study and revise the methods section.
- For some topics we increased the number of citations to provide a more comprehensive representation of different perspectives in each area.
- As mentioned earlier, we emphasised that this collection does not claim to be comprehensive and absolute. Rather, it is our intention to publish this type of work on a regular basis, ideally annually.

5. Alternative to all the above – and this would involve a massive overhaul of the manuscript – I feel the 9 sections could actually be supplementary information themselves, and some sort of synthetic distillation of everything in a much shorter manuscript could be written. This would likely provide a much more impactful and meaningful article. You might also consider adding conceptual figures or tables to better convey the information in a much more concise and digestible fashion.

As this project is planned to be repeated annually, we will consider these conceptual ideas for the next paper in this series, but for the current manuscript we will skip this alternative route in order to publish the paper as soon as possible. Thank you very much.

There are also some instances of oversimplification of topics that have the potential to be misused in the recommendations (e.g., "NCS should prioritize reforestation of mangroves"). I work extensively in mangroves and am very partial to these ecosystems – but the field generally believes that conservation of existing natural forests should be prioritized before reforestation; and who is to say that mangroves should be prioritized over other ecosystem

types?

We adapted this statement and carefully reviewed others.

It's also worth noting that there are extensive typos and many instances of awkward grammar. Given the accomplishments of the author team, I expected more professionalism in the preparation of the manuscript.

We carefully edited the manuscript to remove awkward grammar and typos.

We also revised the text based on your helpful and constructive minor suggestions. Thank you for your assistance.

Friedrich Bohn on behalf of all Co-Authors.

**Minor suggestions:**

Line 28 – some awkward language; reads as though IPCC & IPBES are "international negotiations," which they aren't.

we reformulated the sentence to omit misunderstanding.

Line 40 – There is a working assumption that IPCC and IPBES are what those implementing biosphere stewardship primarily respond to. Do you have any evidence for that? There are also many other scientific publications that are released (e.g., the annual Global Carbon Budgets) and many land management agencies I think are less tied to these high level reports (thinking of the US Forest Service, which uses national level data and policies).

We have revised the section where other more frequent publications are listed. We have added national reports to this section. In addition, we have shortened our argument here and referred to the large number of other publications in the following section.

Line 47 – There is also the "scientists' warnings" series / community.

We have included the scientists' warning series.

Line 53 – How are you defining "biosphere"? IPCC and IPBES include many scenarios that incorporate social and economic trajectories. The planetary thresholds group out of PIK also does lots of similarly themed research, so I'm not sure I buy the idea that this is unaddressed.

We have reformulated this sentence (and paragraph) and added a definition of 'biosphere' for this paper. We have also mentioned the interdisciplinary approaches of other reports (e.g. IPCC and IPBES) where they are mentioned in the manuscript.

Line 68 – Doing my duty as a critical reviewer … I am not sure any of this is novel or an emerging theme. I think it's clear that biodiversity loss, land degradation, chemical pollution, etc., are all interlinked and driven by social and economic systems.

The wording was misleading. We agree that this is not a new insight of this paper. We have reworded the sentence. However, this general insight should be communicated again and

again, especially in papers that are not exclusively aimed at the scientific community, as this knowledge is not yet taken into account in all decision-making processes.

Line 81 – missing a space before "In the future"

corrected

**Section 3.1**

Section 3.1.3 – I find this section to be rather abstract, with the solutions space disconnected from the challenges. I agree with the statement that inclusive approaches are a must, but think there is a logical disconnect between how community engagement in restoration is likely to alleviate issues such as coastal squeeze.

We reorganized the section and revised the text.

Line 150 – awkward phrasing - "and result in improved"

We reformulated the sentence.

Line 175 –unclear/imprecise. Do you mean physical space? Or metaphorical? Stewardship is governance, so I don't really know what providing hectares for governance means?
As recommendation sections are removed, this item is no longer part of the manuscript.

Line 177 – Indigenous is typically capitalized as a sign of respect; would recommend this throughout your manuscript.

We agree and checked.

Line 181 – This statement is very problematic and has great potential to be misused. When designing NBS for CC, there are many other solutions that should be prioritized over mangrove reforestation (e.g., conservation of existing biodiverse mangroves, or providing them accommodation space). I agree with the article's main argument (that mangrove reforestation is more effective than afforestation), but this bullet point does not capture that at all. Suggest deleting or clarifying.

As recommendation sections are removed, this item is no longer part of the manuscript. However, the sequestration rate of mangroves is added to the background section.

**Section 3.2**

Line 222 – Awkward grammar in this paragraph.

we revised the full paragraph

Line 241 – empty parentheses

removed

Section 3.2.3 – Is restoring forests, which takes many decades and is extremely limited in terms of spatial footprint, really the solution to the issue of a positive feedback loop between deforestation & increased drought-driven forest loss / degradation? To me, the solution is to halt deforestation before you think about reforestation. The best way to do this is likely through strengthened environmental governance, whether through policies, monitoring, enforcement, more resources, or voting folks into power that care about the issue. You might consider whether reforestation and afforestation should be the focus of this section – or whether it should be more focused on strengthening efforts to halt deforestation (is there any evidence to show restored forests help with water cycling in expansive tropical forests)?

We agree and emphasised the need to stop deforestation in a new paragraph and shorten the other paragraphs.

Line 243 – citation? Is this based on modeling in which you have fully restored forests, or empirical data in which forests take many decades to recover structure and canopy cover?

Deleted because of shorten the paragraph

Line 265 – So protecting Amazonian forests from deforestation is not a priority? We have seen massive droughts across the Amazon this year, might consider rephrasing this.

As recommendation sections are removed, this item is no longer part of the manuscript.

Line 275 – Critical in what sense?

As recommendation sections are removed, this item is no longer part of the manuscript.

**Section 3.3**

Line 285 – pulling in opposite directions but regionally distinct, correct? Or are these patterns generally occurring in the same locations?

Human activities and meteorological fire danger can alter fire regimes globally, both in distinct regions and within the same area. For instance in Spain, burned area has decreased for several years even though meteorological fire danger increased. One study suggested that this decrease is attributed to human activities such as improved fire suppression resources and land-use changes (see https://doi.org/10.1007/s13595-019-0874-3). We adjusted the revised text by adding "Across the globe the two factors may change individually or in conjunction", for better clarification (L285).

Line 291 – Interesting – this is a surprising result to me, particularly given the massively extensive fires in the boreal region last year. Also occurrence is rather imprecise. Do you mean extent of fire? Or frequency? Is the general pattern that we are getting fewer but more intense fires across the globe? And this is specific to forests? Or grasslands as well?

The term "fire occurrence" was indeed misleading in this context. In this sentence, we are specifically referring to trends in burned area (as described in the sentences before). Globally, burned area has declined by 1.21 ± 0.66% per year over the past 20 years (Chen, 2023), including the latest estimates for 2024 from GWIS. This decline is primarily driven by reductions in fires in savannas, grasslands, and croplands, as mentioned earlier in this paragraph. We have updated the terminology to "burned area" to ensure consistency—thank you for pointing out this discrepancy.

Megafires appear to be an emerging phenomenon (see the interesting UNEP site), particularly in forest regions of the extratropics, where vast amounts of carbon are emitted into the atmosphere, as recently discussed in Jones (2024). However, to date our current understanding of extreme fires and megafires is still limited.

UNEP https://www.unep.org/news-and-stories/story/are-megafires-new-normal

Jones (2024) https://www.science.org/doi/10.1126/science.adl5889

Line 318 – This pattern of reduced area of burns with increasing fire conditions sounds like what the US implemented for 100 years and is part of what is now driving catastrophic high-severity fires. I.e., you don't burn areas that normally burn, fuels build up over decades, and then the climate gets to a point such that all of your accumulated fuels go up in flames at once. There are many fire studies that argue we should rethink our relationship with fire, and start with the fire ecology of the system. (i.e., suppression might not be a great solution).

We absolutely agree, we also believe that suppression efforts are not sustainable over the long term. We now state this in the revised text by "Moreover, fire suppression should be limited in areas where regular low-intensity fires play a vital role in naturally clearing fuels. There, maintaining fires as a part of the ecosystem can reduce the risk of more severe fires from excessive fuel accumulation."

Line 322 – The U.S. is being MASSIVELY impacted by severe fires – surprised that you find economically developed societies are less affected.

Yes, for the sake of brevity we oversimplified here and actually missed a lot of important nuance. We suggest this more complete text.

"Several studies emphasize that burned area is negatively related to the Human Development Index at both global \citep{chuvieco_human_2021, teixeira_representing_2023} and continental scale (Forrest et al. in press). This demonstrates that more economically developed societies tend to reduce their burnt area, either due to effective fire prevention measures or because of rapid and successful firefighting (see also Section \ref{sec:global}). Whilst this broad picture is encouraging, it is important that this view is tempered with knowledge that relying on fire suppression as a sole strategy is risky and potentially counterproductive, as it can increase fuel accumulation and therefore fire severity (Kreider et al. 2024). A clear example of this is the forests of the United States where, despite a high level of economic development, burnt area is increasing (Iglesias et al 2022, Chen et al 2024, Table 4). Whilst climate change plays a large role in this trend (Iglesias et al. 2022), a very effective strategy of fire suppression over the 20th century (Magerl et al. 2022) without a sufficient fuel reduction strategy has led to current levels of very high fuel accumulation. These high fuel loads are contributing to the current crisis, a phenomenon that was anticipated over 50 years ago (Dodge 1972).

Line 325 – Why not start this solution paragraph with recommending developing an understanding of the fire ecology of the system, and locally adapting strategies and tools for fire management to that system (whether they be suppression, Rx fire, forest types, etc.).

Agreed. Unfortunately, this section is not included in the revised manuscript.

Section 3.3.4 – These are good recommendations. Thank you.

Added references :

Dodge, M.: Forest Fuel Accumulation—A Growing Problem, Science, 177, 139–142, https://doi.org/10.1126/science.177.4044.139, 1972.

Forrest, M., Hetzer, J., Billing, M., Bowring, S. P. K., Kosczor, E., Oberhagemann, L., Perkins, O., Warren, D., Arrogante-Funes, F., Thonicke, K., and Hickler, T.: Understanding and simulating cropland and non-cropland burning in Europe using the BASE (Burnt Area Simulator for Europe) model, Biogeosciences, in press, https://doi.org/10.5194/egusphere-2024-1973, 2024.

Iglesias, V., Balch, J. K., and Travis, W. R.: U.S. fires became larger, more frequent, and more widespread in the 2000s, Science Advances, 8, eabc0020, https://doi.org/10.1126/sciadv.abc0020, 2022.

Kreider, M. R., Higuera, P. E., Parks, S. A., Rice, W. L., White, N., and Larson, A. J.: Fire suppression makes wildfires more severe and accentuates impacts of climate change and fuel accumulation, Nat Commun, 15, 2412, https://doi.org/10.1038/s41467-024-46702-0, 2024.

Magerl, A., Gingrich, S., Matej, S., Cunfer, G., Forrest, M., Lauk, C., Schlaffer, S., Weidinger, F., Yuskiw, C., and Erb, K.-H.: The Role of Wildfires in the Interplay of Forest Carbon Stocks and Wood Harvest in the Contiguous United States During the 20th Century, Global Biogeochemical Cycles, 37, e2023GB007813, https://doi.org/10.1029/2023GB007813, 2023.

**Section 3.4.**

Line 359 – This might be misleading. Terrestrial ecosystems have absorbed a roughly consistent proportion of our annual emissions for several decades but it is due to a mixture of effects – CO2 fertilization is a big one, but also lengthened growing seasons in northern latitudes, and regrowth of forests in northern / developed countries have contributed.

We agree, these other processes, though CO2 fertilization remains the dominant driver of terrestrial carbon uptake. We reformulate the text.

I wouldn't note the Co2 fertilization effect unless you are talking about the fact that terrestrial ecosystems have uptaken roughly the same percentage of our annual emissions over many decades.

We clarified in the text that we are indeed talking about sustained decadal-scale uptake

I also wouldn't use "additional" removal as additionality has a very specific meaning within Nature-based CDR.
We reformulated the sentence to clarify that we are not referring to additionality here

Happy to mention t

Line 3.4.3. – There are many critiques of tonne-year accounting, and also many other potential solutions to NCS implementation (citing projects in areas where disturbance is less likely, designing solutions to confer resilience to ecosystems, focusing on strategies that might be less prone to disturbance such as agroforestry, etc.). This section feels cursory and under-developed relative to the others. I am also not sure that treating NCS as temporary resolves all implementation risks (thinking about issues related to additionality, leakage, environmental justice, etc. – it only addresses the permanence criteria).

We added text to clarify that the proposed model is a different application of tonne-year accounting (which also critiques previous applications)

**Section 3.5.**

This section is well-developed and the logic is consistently threaded through the various sub-sections nicely (background – challenges – solutions – recommendations). In the interest of pushing the authors further, wouldn't there be unique challenges to restoring 25-30% of every developed square kilometer (this seems massively expensive). What are those social and economic barriers, and do you have concrete ideas for how to work past them?

Thanks for pointing this out. We agree that restoring 20-25% of every developed square kilometer presents unique social and economic challenges. We have incorporated your suggestion into the text by mentioning key barriers, such as high restoration costs, land tenure issues, policy constraints, lack of expertise and knowledge and potential conflicts with land uses like food production and housing. In the solutions sub-section, we also added a sentence emphasizing adaptive, context-specific approaches that focus on prioritizing areas where habitat restoration can align with community needs, supporting both biodiversity and socio-economic priorities.

**Section 3.6**

**This section is completely rewritten**

This feels like a large enough topic to encompass all other topics covered in this manuscript – is there a way to make this section more focused?

Lines 495-510 – I suggest deleting or condensing this. This section feels longer than the others, and I don't think this broad framing is needed given the article's intended audience.

Line 540 – Would this paragraph be better placed in solutions? E.g., reframe policies section to "design policies to transition to a circular economy"

Line 565 – I really like these examples as they provide concrete guidance on policy recommendations, which is lacking in many other sections. However, some of this can be condensed (e.g., you already describe the CBD and other agreements in the intro). Consider focusing the sentence more towards solutions (e.g., keep the sentences on how international agreements can be improved).

Line 605 – I like these recommendations. However, the author team should align the recommendations. Most are just a line or two, whereas these are fleshed out with examples. You might consider condensing these down to pithy recommendations.

**Section 3.7**

Line 630 – Could delete or rewrite the boilerplate first sentence.

Deleted.

Line 635 – Love that you are bringing political ecology into this. So critical for many of the issues that are highlighted in this article. I think it also adds credence to the interdisciplinarity claims of the article.

Thank you, and yes we agree.

**Discussion**

I am not sure how useful the 2-3 sentence highlights of the 9 topics actually is. Surely, given that you are focused on interconnections across biosphere stewardship, there are synergies across the nine sections. What are those? Can you provide us with some sort of synthetic view or key themes that emerge across all of the areas? This would be most helpful for the policy-makers, environmental stewards, and researchers that you suggest are the audience of this article.

We completely rewrote the first part of the discussion and developed 4 synthetic findings (each with two to three sub-points in bold) on the 8 themes. We hope this section is now much more helpful for the target group of this paper

---

## Author Response (AR2)

**Review #1**

This manuscript is now much more streamlined, clear, and easy to follow. I appreciate the authors' responses to my comments and think they have done a nice job of restructuring the manuscript sections. I especially liked the identification of the 4 summary ideas at the end. This section really highlights the importance of the review and nicely synthesizes challenges and next steps regarding our knowledge of and interaction with Earth's biosphere. The updated introduction also did a good job of describing the importance of the synthesis up front, setting up a stronger structure for the reader to follow. I did notice that a number of grammatical/punctuation errors remained--it looks like this may have been the case where sentences were heavily edited and merged (also a missing reference at line 520). I would recommend a technical double check before the final publication, but I have no remaining substantial edits to suggest. This manuscript makes a nice contribution to advancing studies of the biosphere!

Thank you very much for your thoughtful and encouraging feedback. We are pleased to hear that the summary section effectively summarises the challenges and next steps in biosphere research. We have revised the manuscript to ensure that grammatical/punctuation errors are addressed before final publication.

**Review #3**

Thank you for your thoughtful and constructive feedback on our report. We greatly appreciate the time and effort you've invested in providing such detailed recommendations.

The title is broad and appropriate for the review's scope. However, it could be more specific to highlight key focus areas, such as climate change, biodiversity, and socio-economic interactions. This would better capture the central themes of the paper and attract more readers.

Good point: We update the title by giving a subtitle: "Reviews and syntheses: Current perspectives on biosphere research - 2024:  8 findings from ecology, sociology and economics."

To improve clarity an**d reader engagement, it might help to include a sentence summarizing the eight selected themes in the abstract.**

Highlighting key actionable insights in the abstract would emphasize the review's contributions.

The abstract covers the scope effectively but could benefit from a more concise and impactful structure. A better approach might be to introduce the paper's aim as an interim report addressing the biosphere crisis for policymakers and stakeholders right at the outset. This quickly establishes the purpose. Again, introducing the eight themes briefly in the abstract would provide readers with a quick overview of the review's breadth.

We have revised the abstract to improve its structure by including actionable insights and using the same terminology as in the paper.

The introduction is somewhat long-winded and takes time to introduce the objectives and chosen methodology. Some suggestions for readability:

1.      Begin with the document's objective as an interim report for policymakers and stakeholders.

2.      Clearly outline the purpose of addressing challenges and bridging knowledge gaps between IPCC/IPBES reports.

3.      Include a brief mention of the eight themes/topics here, with consistent references to these themes throughout the abstract, introduction, and conclusion. This consistency will improve the narrative flow and reader comprehension.

Thank you very much for these helpful comments:

- We have revised the introduction to make the aim of this (and subsequent papers in this series) clear in the first section.
- We have slightly shortened the introduction.
- The new text emphasises more strongly the bridging nature of this review.
- We have carefully revised the last section to make it consistent with all parts of the paper.

The background sections are informative and relevant but could be tightened for precision. Use more specific data (percentages, years, quantitative findings) to support the points. For example, in some areas, numbers are presented effectively, while others rely on broad statements.

In the backgrounds, avoid generic reiterations of the biosphere's importance. Instead, tie the relevance of background information directly to the specific topic of the sub-section. This will reduce redundancy and improve focus.

Thank you for your suggestions. We have selectively revised the background sections.

In section 3.2, we have added some figures to quantify the water flows

In section 3.3 we have also added some specific data to provide an overview of recent fires in different regions of the world.

In Section 3.5 we have added some data documenting the decline in biodiversity and the proportion of land under protection. We further have added the dependence of crops on pollinators as an example of the importance of biodiversity for human well-being.

In section 3.6 we have added here as well more specific data related to the specific topic.

The challenges are well-documented, but the implications need to be tied more explicitly to human lives, biodiversity, and socio-economic systems. For instance, "500 million people are projected to experience challenges within decades due to the likely loss and degradation of coral reefs" should specify what challenges people face and why they rely on coral reefs.

Adding such detail will make the stakes clearer and more impactful. This recurring issue in the manuscript should be addressed across all sections.

Thank you for your helpful feedback, we have added more explicit details about the impacts on human life, biodiversity and socio-economic systems in selected sections:

in Section 3.1. we added specifics regarding the how and why each statement was made in both the challenges and solutions sections as appropriate.

in Section 3.2. we add a short section about social economic loss due to droughts.

In Section 3.3. We explain in more detail the challenges for society, forestry and civil protection.

In Section 3.6. We added more data to show the linkages between human lives, biodiversity and economy.

The solutions are relevant but lack sufficient detail to make them practical. For example, "Better allowing space for stewardship practices by Indigenous and local communities can provide meaningful lessons..." is valid but vague. What specific actions, policies, or mechanisms should be implemented to allow for stewardship practices? Adding such specificity will help the paper achieve its intended impact.
•         Similarly, it would help to add examples of successful implementations (e.g., case studies, projects) where relevant.
•         There is an acknowledgement of biases, but the explanation is insufficient. For instance, biases toward tropical forests in the section on forest protection should be explicitly discussed. What is the basis for this focus? Are there overlooked ecosystems that also deserve attention? Addressing such caveats would strengthen the paper's credibility and transparency.

Thank you very much for your suggestions to improve solution sections:

For section 3.1, we have made more additional mention of local examples (e.g."Bakhawan Mangrove Eco-Park" in the Philippines) and restructured to highlight benefits of solutions both locally and globally in relation to the challenges section.

For section 3.2, we note that other systems beyond the tropical forests were already discussed (savannahs, grasslands). Nevertheless, we reworded the last paragraph of the section to more explicitly include the need for spatially distributed observations.

In section 3.5 we have added a recent study showing that 20% natural habitat is important to maintain natural pollination.

In Section 3.8, we added a few lines and literature about Conservation Basic Income (CBI) initiatives that are currently taking place and about bottom up cohabitation with animals, specifically highlighting human-bear relations in Bulgaria.

The conclusion does a good job of summarizing findings, but it could more strongly articulate the report's overarching goal as a call to action. Emphasize:

•	The collaborative methodology of bringing together experts across disciplines and geographies.
•	The unique contribution of this review in providing timely, actionable insights as an interim measure between major IPCC/IPBES reports.
•	The importance of addressing interconnected crises (climate, biodiversity, socio-economic systems) through interdisciplinary and inclusive approaches.

Thank you very much for your suggestions to strengthen the conclusion. We revised the conclusion and
- highlight the collaborative, interdisciplinary nature of our methodology and the importance of addressing the interconnected crises of the earth system.
- We also underscore the unique contribution of this review in bridging the gap between major IPCC/IPBES reports.

General Recommendations:
o	Reduce redundancy by removing sweeping, generic statements.
o	Ensure grammatical and formatting errors are corrected.
o	Break up longer paragraphs for better readability.
o	Provide specific, actionable recommendations in the solutions sections.
o	Focus on making the findings practically relevant for policymakers and stakeholders.
o	Reinforce the unique aim of this review: synthesizing diverse perspectives and expertise to address the Earth system crisis.
o	Use consistent terminology and thematic references throughout the document for better readability and coherence.

Thank you also for these recommendations. We worked on reducing redundancies, correcting grammatical and formatting errors, improving paragraph structure, using consistent terminology,  revising some of the solution sections, and highlighting the diversity of disciplines contributing to this paper. Finally, we have checked for consistency of terminology and references. There will also be  a homepage which presents the actional recommendations of the solution sections in a compact way.

Thank you again, your comments have contributed significantly to improving the quality of the paper. We're grateful for your expertise and the time you've dedicated to this review process.

---

## Author Response (AR3)

Dear Reviewer 4,

Thank you for your helpful feedback. We greatly appreciate the time and effort you have put into improving the manuscript.

Usage errors remain, e.g. 'we synthesis four' in the abstract. Please re-read carefully for content e.g. lack of space on line 164, 170 etc.

Thank you for pointing this out. We have corrected the tense and reread the paper carefully to check for missing spaces.

We are also making the same small additional changes in grammar and wording: e.g. updated the title, as we have 2025 and some references date from 2025.

We also use topics, themes, findings and overarching insights consistently:
Several **topic** are suggested and collected by the survey. The editorial board selected 8 **themes** (e.g by merging, extend and rejecting topics), which include "new" **findings** of research (beside well known "old" background information). By analysing the themes, we have synthesised 4 **overarching insights**.

See all tiny changes in the track changes document.

30: what is intended for the 'series'? perhaps just note it is a report as its unclear what will happen going forward although I agree that the Reviews and Syntheses format in Biogeosciences is a good way to do so.

Good Point.We rephrase the corresponding sentence to make the intention of the series clearer

I respect the honesty of fig. 1 and the imbalance in responses but this figure could be improved by including numbers in addition to the pie chart.

Done

103: who is the editorial board?
We revised the sentence to better describe the editorial board: The editorial board, consisting of six professors (see authors' contribution) with experience in ecology, sociology and economics, made the final selection based on the following criteria:

123: value should include uncertainty
we added the range of the measurements given in the publication of Alongi 2012

146: quotes aren't needed because it has the effect of trivializing content.
removed

155: 'Stakeholders have insufficiently considered locally relevant species when planning with nature-based solutions' is vague. It helps to avoid the use of 'stakeholder' because this has anti-Indigenous connotations, at least in the U.S.
We reformulate the sentence omiting the term "stakeholders".

250: sentence fragment.
corrected.

3.2.3: some of these arguments are simultaneously oversimplified and incomplete because atmospheric boundary layer dynamics are complex (https://www.nature.com/articles/s41467-021-24551-5), cloud nucleation changes (https://acp.copernicus.org/articles/9/6531/2009/) and much more. I guess that the brief treatment given to each topic is ok if it helps encourage discussion and study.

We agree, as it is always a challenge to simplify without oversimplifying. We have therefore added selected points and refer to the relevant papers.

I'm not entirely convinced that section 3.8 is a topic but may be part of solutions.
We have chosen this topic to inspire and motivate the reader to think outside the box, especially in the long term.

Thank you once more for your helpful feedback!